

# The sagging shape of shoreline formed on downdrift side of the structures due to seasonal oblique wave incidence

Changbin Lim[1], Soonmi Hwang[2], Jung Lyul Lee[3]

[1]School of Civil, Architecture and Environmental System Engineering, Sungkyunkwan University, 2066, Seobu-ro, Suwon
16419, Korea
[2]Department of Coastal Management, GeoSystem Research Corporation, 172 LS-ro, Gunpo 15807, Korea
[3]Graduate School of Water Resources, Sungkyunkwan University, 2066, Seobu-ro, Suwon 16419, Korea

*Correspondence to*: Jung Lyul Lee (jllee6359@hanmail.net)

**Abstract.** Downdrift coastal erosion has occurred at natural or man-made groynes on Korea's eastern coast, caused by oblique

high waves in winter months. The resulting shoreline planform has a sagging shape with a maximum indentation point within

the eroded shoreline. This study focused on solving the frequent and severe coastal erosion problem of this type at the

Jeongdongjin review of wave data over 40 years from the National Oceanic and Atmospheric Administration (NOAA), as well

as analyzing shoreline monitoring images for identifying the yielding line of maximum indentation points. An analytical

method was developed to verify the eroding shoreline in a sagging shape and its maximum indentation by applying the

conservation principle of sediment transport and the empirical model of equilibrium shoreline. To examine how well the

empirical formula is suitable for the Jeongdongjin coast, the annual directional spectrum of the observed wave data was applied

to the simple diffraction wave model for the gamma breakwater, and satisfactory agreement was obtained by comparing it with

the shoreline results. Breaking wave height and angle, duration, longshore sediment transport coefficient, and protruding length

of the groyne were the inputs. The theoretical results are in good agreement with those of the shoreline monitoring program.

The factors mitigating downdrift coastal erosion of this type were identified by applying the obtained theoretical solution, and

the engineering solutions were examined via factor analysis.

## 1 Introduction

On the eastern coast of Korea, low waves from the ENE direction prevail in summer, whereas high waves from the NE direction

dominate in winter. Thus, in areas where seasonal wave direction changes, local shoreline orientation responds to changes in

wave direction (for example, Sokcho and Jeongdongjin Beaches in Korea; Kim and Lee, 2015). Recently, coastal erosion of

more than 30-m in length has occurred often along the eastern coast of Korea because of seasonal high waves at the downdrift

of natural groynes. A typical case of erosion of this type occurred in 2016 at downdrift of the northern groyne in Sokcho Beach

(128o36'14"E, 38o11'25"N) in Gangwon-do Province, caused by winter high waves. Since then, a shoreline maintenance

project has been implemented, with an extension to the groynes and four submerged breakwaters. In addition, erosion of

approximately 40 m was observed in 2016 at the updrift of Jeongdongjin Beach (129 o02'26"E, 37 o41'37"N) in Gangwon-





do, where oblique high waves in winter encountered a group of natural (pillar) rocks protruding 80 m into the open sea, damaging rail-bike (pedal-powered rail-cycle) tracks and the inner wall of the Hourglass Park, as shown in Fig. 1.

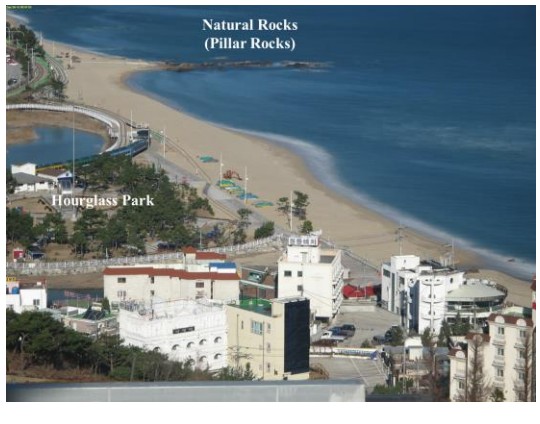

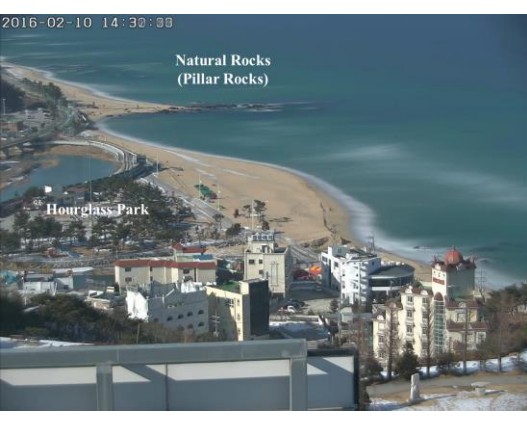

(a)                                                                                  (b)

**Figure 1: Erosion caused by seasonal high waves at Jeongdongjin Beach, Korea: (a) normal wave on December 04, 2015; (b) high wave on February 10, 2016.**

Shoreline planforms are either straight or curved, resulting from incident wave action on geologically inherited coasts (Inman and Nordstrom, 1971). Many studies have investigated the erosion on the downdrift side of structures. A shoreline's long-term stability depends on the balance between the longshore sediment transport (LST) in a coastal environment and sediment characteristics at wave breaking (Higgins, 1970; Komar and Inman, 1970; USACE, 1984; Kamphuis, 1991; Bayram et al., 2007). Among the available empirical models—such as the logarithmic spiral (Yasso, 1965), parabolic bay shape equation

(PBSE, Hsu and Evans, 1989), and hyperbolic tangent model (Moreno and Kraus, 1999)—only the PBSE applies to the static equilibrium planform (SEP) in the vicinity of or bounded between impermeable headlands (structures). On a beach with SEP, the LST becomes static to maintain the shoreline planform under the same predominant wave direction (Silvester and Hsu, 1993). Together with the equilibrium beach profile, the concept of SEP has been widely used as an engineering design tool for beach nourishment projects (González et al., 2010), as well as a means for project planning. Recently, Lim et al. (2019) also

demonstrated the validity of the PBSE using wave data from the eastern coast of Korea to estimate the impact of engineering structures (for example, jetties and groynes).

To reproduce the shapes of shorelines in the laboratory, Badiei et al. (1995) conducted physical experiments on topographic changes to measure the effect of a groyne, and Leont'yev (1997) proposed a short-term shoreline change model associated with groyne-type structures. In addition, Wang and Kraus (2004) performed laboratory experiments on shoreline changes

induced by groynes, which blocks LST. As a numerical approach, Pelnard-Considere (1957) proposed a one-line model that can simulate the temporal changes of a shoreline following the construction of a groyne on a beach. The applicability of this model has been verified in various situations by applying the concept of longshore diffusivity (for example, Le Mehaute and Soldate, 1979; Walton and Chiu, 1979; Larson et al., 1987). Among them, Ozasa and Brampton (1980) developed a beach topography change model that can simulate the sagging shape of the shoreline due to wave diffraction. More recently, Lim et





al. (2021) applied the parabolic bay shape equation (PBSE or parabolic model) of Hsu and Evans (1989) to indirectly reflect the effect of wave diffraction caused by coastal structures. High-resolution numerical models composed of waves, currents, and topography change modes have also been developed to simulate topographic changes around structures (for example, Xbeach and Sbeach). These laboratory experiments and numerical modeling contributed to solving scientific questions about downdrift erosion by providing a similar sagging shape. However, engineering countermeasures to reduce coastal erosion

damage on the downdrift side of structures still lack reliable factor analysis.

This study developed a theoretical approach, using the PBSE, that can predict the critical point (i.e., maximum indentation) in a sagging shoreline planform on the downdrift side of an LST barrier. The validity of this approach was examined by comparing its results with those from a shoreline change model that converges to the equilibrium shoreline of the SEP. Further, as the method was verified using the video monitoring data at Jeongdongjin Beach, where sagging shoreline curves have formed

frequently by seasonally changing waves at the downdrift of a group of large irregular natural rocks, as shown in Fig. 2.

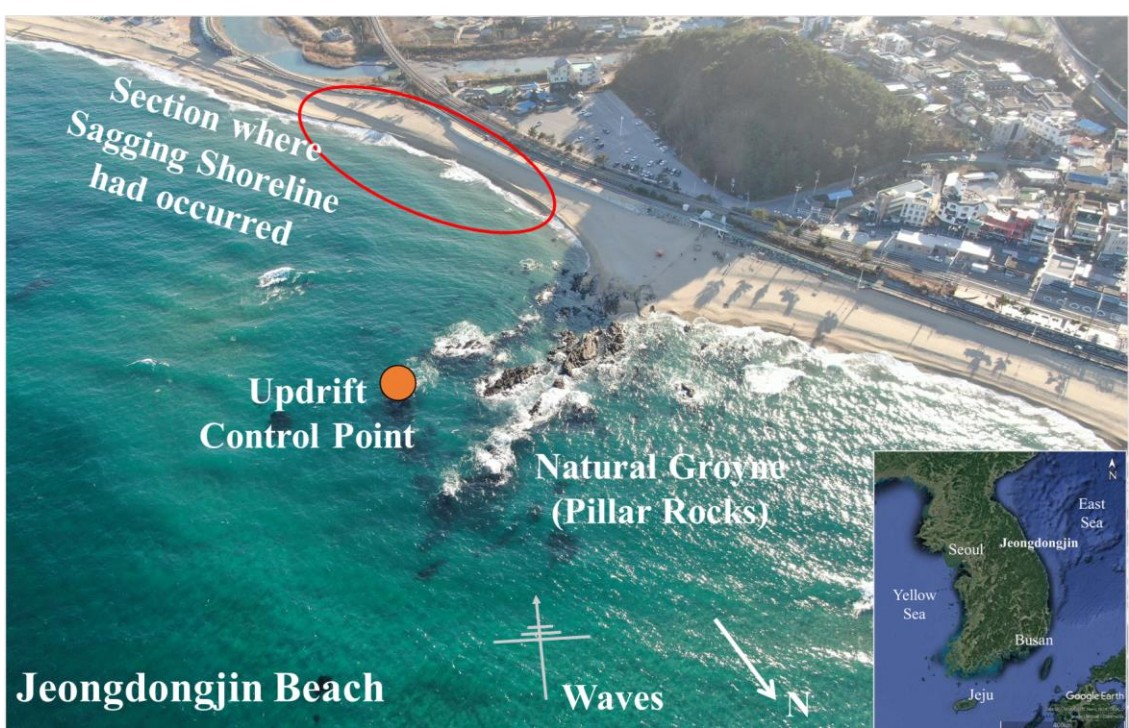

**Figure 2: Location of pillar rocks seaward of Jeongdongjin Beach, where sagging shoreline planform had occurred © Google Earth 2021.**

Section 2 describes the analysis of seasonal wave distribution from NOAA's wave data and the longshore sediment transport

equation. Section 3 presents the well-known parabolic model (Hsu and Evans, 1989) for a stable bay shape and its approximation for the theoretical analysis of the eroded shoreline planform, which appears as a sagging shape with a maximum indentation point between the updrift and downdrift control points. A theoretical solution for the downdrift control point on the eroded shoreline is proposed in Section 4, together with that for locating the maximum indentation. These results are then



compared with the numerical solutions in Section 5 and with the recorded images obtained from the shoreline monitoring
program for Jeongdongjin in Section 6. Section 7 presents an engineering response strategy that uses beach nourishment and
groyne facilities, which are representative engineering countermeasures, to solve the coastal erosion problem on the downdrift
side of the structures through a theoretical formula that facilitates factor analysis. Finally, Section 8 provides concluding
remarks.

## 2. Analysis of seasonal incidence waves

### 2.1 Analysis of NOAA data

The European Center for Medium-Range Weather Forecasts and the National Centers for Environmental Prediction (NCEP)
under the National Oceanic and Atmospheric Administration (NOAA) in the United States have provided long-term wave
hindcast data since January 1979. Since 2004, NOAA has also operated the Climate Forecast System Reanalysis and Reforecast
(CFSRR) activity, which analyzes sea climate by using more than 60-years' observation data. Saha et al. (2010, 2014) verified
the usability of NOAA data by assimilating and verifying CFSRR observation data.

Regarding the incident wave conditions in the open sea on Korea's eastern coast, we used 40 years of NOAA data (38.0 °N,
129.5 °E) for Jeongdongjin Beach, represented as $H_i$. Analysis results of the NOAA's monthly wave data were used to simulate
the changes in the sagging shoreline curve around the natural rocky groyne caused by the energy of the oblique high waves.
The monthly root mean square (RMS) wave height, weighted wave period, and weighted offshore wave direction were
calculated using Eqs. (1a)–(1c).

$$\bar{H} = \sqrt{\frac{\sum_{i=1}^{N} H_i^2}{N}} \tag{1a}$$

$$\bar{T} = \frac{\sum_{i=1}^{N} T_i H_i^2}{\sum_{i=1}^{N} H_i^2} \tag{1b}$$

$$\bar{\alpha} = \frac{\sum_{i=1}^{N} \alpha_i H_i^2}{\sum_{i=1}^{N} H_i^2} \tag{1c}$$

where N is the number of wave data, and $\bar{H}$, $\bar{T}$ and $\bar{\alpha}$ are the RMS wave height, period, and direction, respectively. Fig. 3
depicts the monthly variations in the RMS wave height, period, and direction of the significant waves, averaged over 10 year-
intervals from 1979 to 2018. As shown Fig. 1, the NE waves in winter (December–February) arrived at approximately N10°E,
the ENE waves in summer (June–August) approached from N70°E to Korea's eastern coast, and the local shoreline orientation
in Gangwon-do was approximately N43°E.





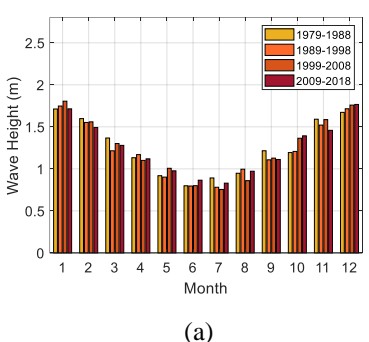 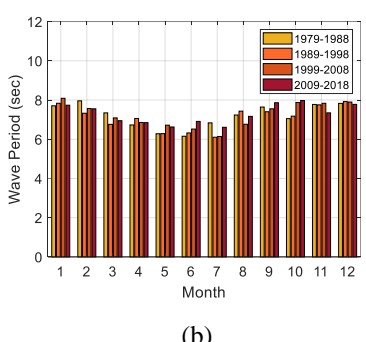 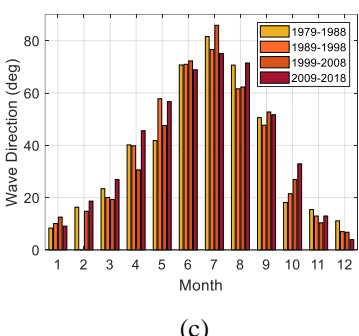

|     |     |     |
| --- | --- | --- |
| (a) | (b) | (c) |

**Figure 3: Monthly wave analysis for the eastern coast of Korea: (a) wave height; (b) wave period; and (c) offshore wave direction.**

## 2.2 Analysis of seasonal longshore sediment transport

Komar and Inman (1970) conducted field experiments on the energy flux in the longshore direction, $P_y$, and the longshore sediment transport (LST) rate, $Q_y$, and reported the following relationship:

$$Q_y = \frac{I_y}{(\rho_s - \rho)(1-p)g} = \frac{KP_y}{(\rho_s - \rho)(1-p)g} \tag{2}$$

where $\rho$, $\rho_s$, $p$, and $g$ are the seawater density, sediment density, sediment porosity (typical approximately 0.3 to 0.4), and acceleration of gravity, respectively. $I_l$ is the immersed weight of the sediment transport rate. $K$ is a dimensionless coefficient (e.g., CERC coefficient) that depends on seabed property and significant wave height, which can be taken as 0.39 (USACE, 1984; but was taken as 0.77 in Komar and Inman). The alongshore component of the energy flux per unit length of beach $P_y$ is defined as

$$P_y = \left(EC_g\right)_b cos\alpha_b sin\alpha_b \tag{3}$$

where subscript b denotes the condition at wave breaking, $\left(EC_g\right)_b$ is the wave energy flux at breaking, and $\alpha_b$ is the breaking wave angle between the shoreline and wave crest line. According to the USACE (1984), sandy longshore sediment serves as a function of wave direction and wave breaking height ($H_b$), as shown in Eq. (4). Here, the unit for the LST rate, $Q_y$, is $m^3/s$.

$$Q_y = CH_b^{5/2}sin2\alpha_b \tag{4}$$

and $C \left(= \frac{K\sqrt{g/\kappa}}{16(s-1)(1-p)}\right)$ has a value of 0.0847 for most types of sand. In Eq. (4), the LST coefficient $K = 0.39$, the acceleration of gravity $g = 9.81 \ m/s^2$, the spilling wave breaking index $\kappa = 0.78$, the sediment specific gravity $s = 2.57$, and the porosity $p = 0.35$ for most types of sand.

In Eq. (4), the LST rate calculated at wave breaking can be expressed by using the deep-water wave data shown in Eq. (5), assuming that the isobath of the seabed is parallel to the straight shoreline. To compensate for the variation in the incident



wave direction by varying the depth change, we modified the superscripts of the wave characteristics by using different
fractional values to calculate the LST rate per unit time:

$$Q_y^O = C_O H_O^{2.4} T_O^{0.2} cos\alpha_O^{1.2} sin\alpha_O \tag{5}$$

where $H_o$ and $T_o$ are the significant wave height and wave period, respectively. The wave direction in the deep water $\alpha_O$ is
measured between the outward normal to the shoreline and wave orthogonal or from the true north direction $\theta_O$, as shown in
Fig. 4. Thus, $\alpha_O = \frac{\pi}{2} - \theta_O + \beta$, with a positive amount of sediment pointing south and a negative pointing north. Furthermore,
$C_O$ is a factor that reflects the characteristics of the sediment and waves, including specific gravity, porosity, breaking index,
and wave angle.

$$C_O = \frac{Kg^{0.6}}{16(s-1)(1-p)(2\pi)^{0.2}\kappa^{0.4}cos\alpha_b^{0.2}} \tag{6}$$

where $C_O$ has a value of 0.0719 approximately for most sands, assuming that the effect of $\alpha_b$ is negligible.

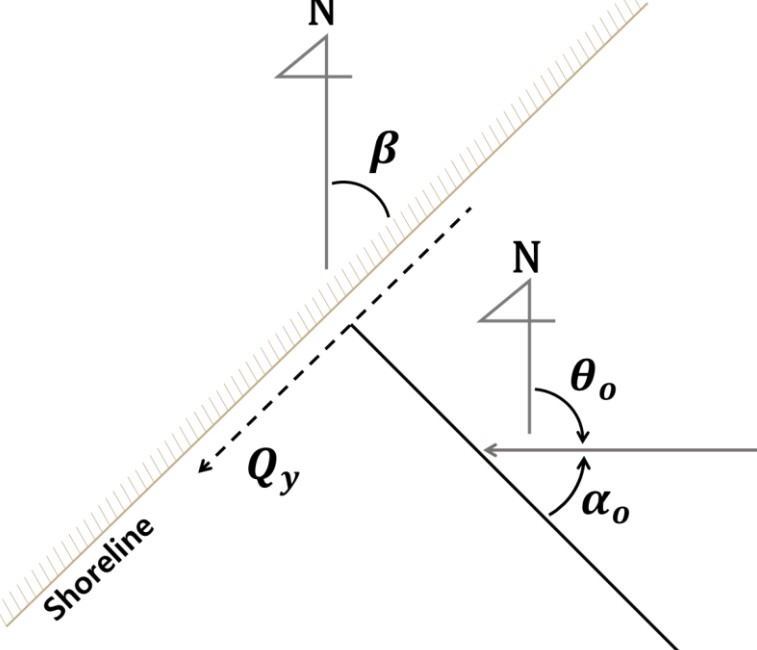

**Figure 4: Definition sketch showing deepwater wave angle $\alpha_O$ with others for calculating longshore sediment transport rate in this study.**

Fig. 5 shows a rose diagram of the annual average incident wave and the LST at the target coast. We calculated the rotation
angle $\gamma$ of the LST increase by adding $\pi$ (180°) to the main wave direction $\theta$. An equilibrium shoreline was achieved when
the northward and southward LST components were equal. If an imbalance of the LST occurs in any season, erosion could
result in shoreline retreat or rotation. The LST obtained from the NOAA wave data is balanced at approximately N38°E,





whereas the shoreline orientation of Jeongdongjin Beach is approximately N43°E, within an allowable error limit in the estimation of wave direction in the open sea and variation in bottom topography.

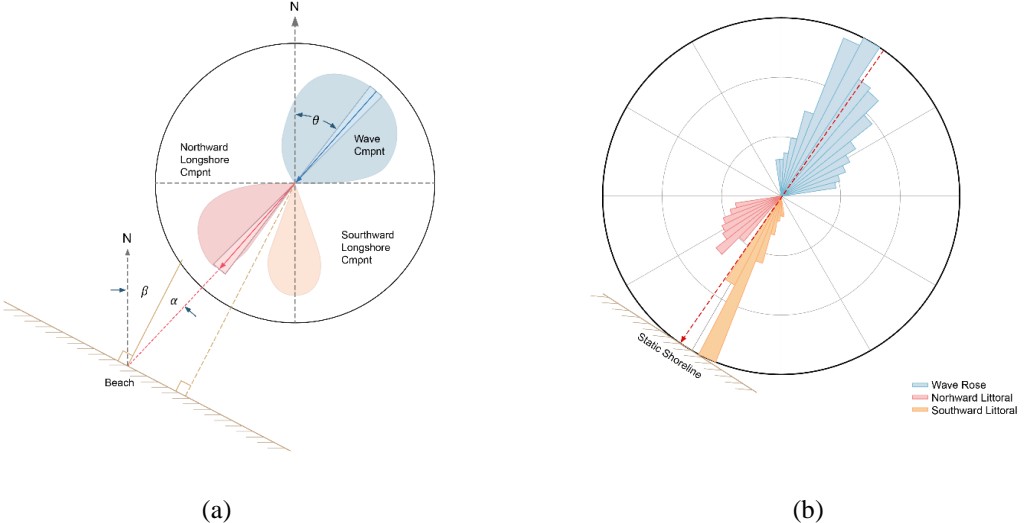

(a)                                    (b)

**Figure 5: Wave and longshore sediment transport directions: (a) definition, (b) rose diagram.**

By using NOAA wave data to plot the monthly wave factor for the LST in terms of $Q_y^o/C_O$, the results shown in Fig. 6 reveal

a strong seasonal-dependent trend in the direction of the LST, highlighting southward transport in winter months (November to February) and northward in summer (July to September). Thus, we can expect coastal erosion with a sagging curve to reach a maximum around February at the end of winter and September at the end of summer. However, if the seasonal LST bypasses the beach without being intercepted by the natural pillar rocks (such as the groyne in Fig. 2) at different water levels and wave conditions, moderate or severe sagging could result.





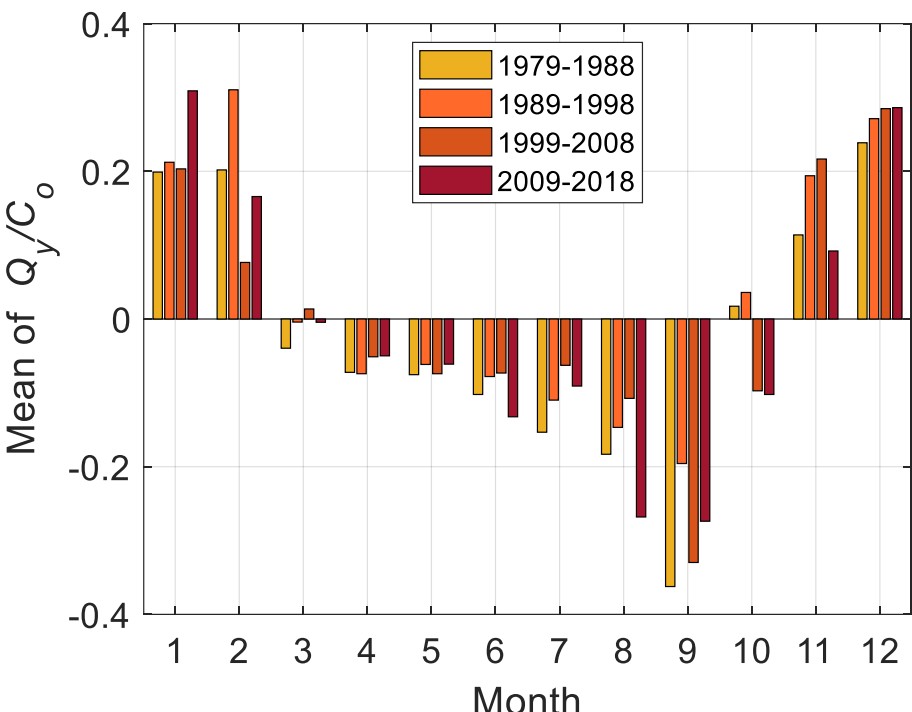


**Figure 6: Monthly wave factor of the longshore sediment transport ($Q_y/C_O$) using NOAA wave data.**

Fig. 7 shows the monthly average of $H_b^{5/2}$ as a function of the oblique wave direction group $\alpha_b$, of 2° intervals at the wave breaking in winter months (November–February) from NOAA wave data. We determined the oblique angle based on the N38°E as $\alpha_b = 0$ and calculated the wave height and breaking angle by N38°E ± 50° in deep water. $H_b^{5/2}$ shows a tendency of

increase or decrease within −7.5° to +12.5°, whereas the number of occurrences exceeded 5000 ranges within −2.5° to +7.5°. These imply that high waves in winter may cause shoreline sagging arrives from the sector within N38°E −2.5° to N38°E +7.5°.



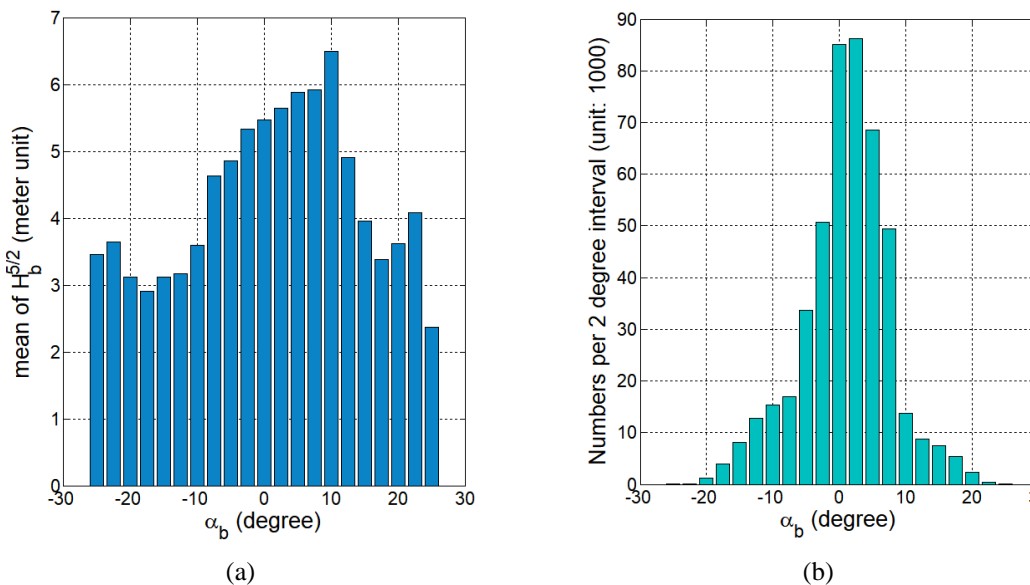

(a)                             (b)

**Figure 7: Analysis of NOAA wave data corresponding to the oblique wave direction group $\alpha_b$ at 2o intervals during winter months: (a) root-mean-square of $H_b^{5/2}$; (b) numbers within each 2° interval (unit: 1,000).**

The LST can be calculated by using Eq. (5) from the waves approaching the shoreline obliquely. However, coastal structures (e.g., natural or artificial groynes) and retention at the updrift coast control the amount of the LST available to a beach. Fig. 8 shows a conceptual sketch of the change in the LST encountering a groyne on a straight shoreline. Before the sediment bypassing the groyne in Fig. 8(a), shoreline advance occurred on the updrift side, with retreat at downdrift. As soon as the sediment bypasses the tip of the groyne, long-term shoreline equilibrium may reach both sides of the groyne. However, during

this process in field conditions, the locations of A and A' may fluctuate, with the former gradually advancing updrift, while the latter slowly shifts downdrift.





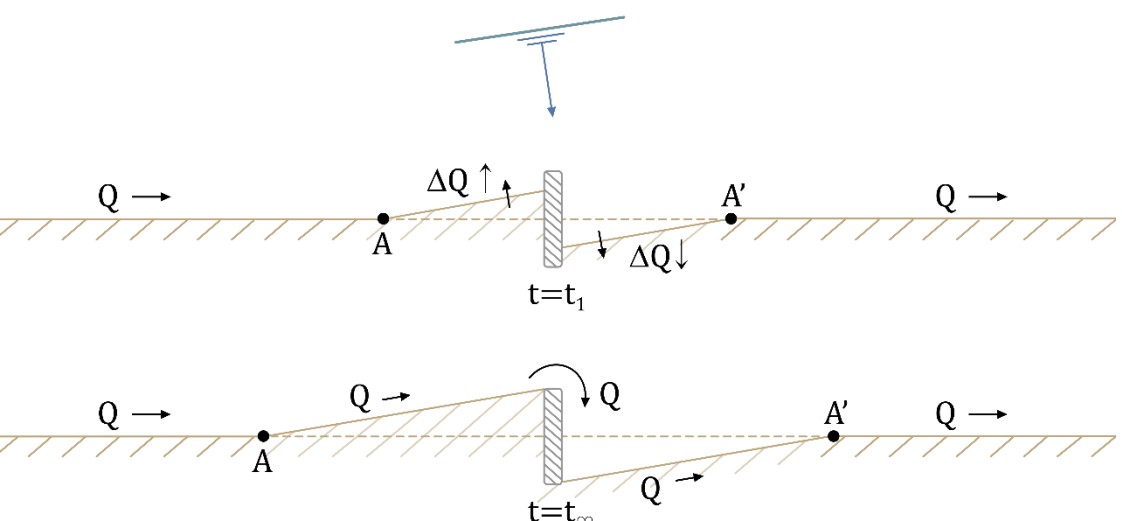

**Figure 8: Shoreline change affected by sediment bypassing a groyne under oblique waves.**

### 3. Parabolic bay shape equation

The PBSE proposed by Hsu and Evans (1989) for embayed coasts in static equilibrium is provided by (Fig. 9)

$$R(\theta) = \frac{a}{sin\beta} \left[ C_0 + C_1 \left( \frac{\beta}{\theta} \right) + C_2 \left( \frac{\beta}{\theta} \right)^2 \right] \ for \ \theta \geq \beta \tag{7a}$$

$$R(\theta) = \frac{a}{sin\beta} \ for \ \theta \leq \beta \tag{7b}$$

in which $R$ represents the radial distance from the parabolic focus to the point on the equilibrium shoreline; $a$ is the distance between the wave crest line (wave crest base line) and the line that passes through the downdrift control point parallel to the

shore base line; $\beta$ is the angle between the wave crest base line and the line joining the focus and the downdrift control point; $\theta$ is between the wave crest base line and the radius R for the point on the equilibrium shoreline, and $C_0$, $C_1$, and $C_2$ are the coefficients derived from regression analysis for static bay shapes (Hsu and Evans, 1989). At the downdrift control point, the boundary condition $C_0 + C_1 + C_2 = 0$ is satisfied to endure a common tangent at $\theta = \beta$.





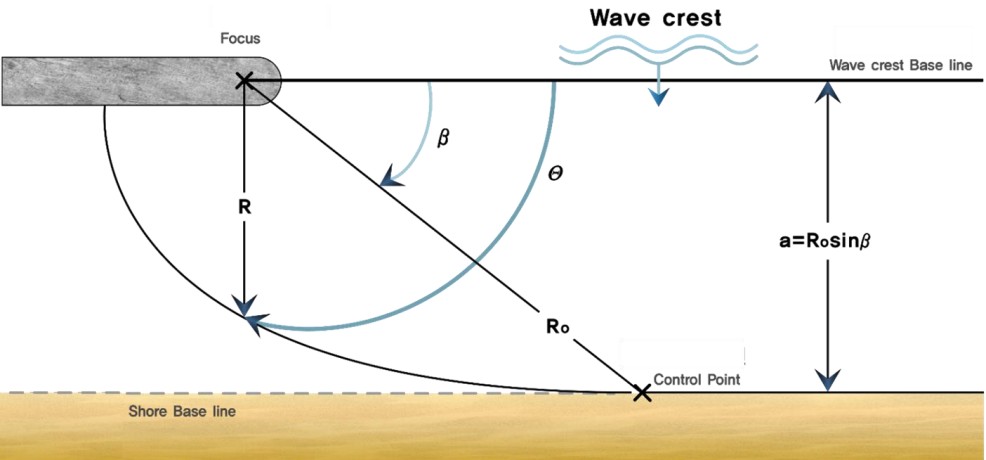

**Figure 9: Definition sketch of parabolic model.**

When the downdrift straight section of an embayment is long, Eq. (7a) can be approximated as

$$R(\theta) = \frac{\beta}{\sin\beta} \frac{a}{\theta} \tag{8}$$

Within this stretch, waves break at an angle $\alpha_b$, which can be expressed by Eq. (9) from the approximation of Eq. (8). $\beta$ absent from in Eq. (9) implies that $\theta$ can be solely determined from the wave angle $\alpha_b$ at wave breaking, or vice versa.

$$\alpha_b(\theta) = tan^{-1}\left(\frac{sin\theta - \theta cos\theta}{cos\theta + \theta sin\theta}\right) \tag{9}$$

Table 1 and Fig. 10 show the relationship between $\theta$ and $\alpha_b$ in Eq. (9) for the equilibrium shoreline.

**Table 1: Relationship between oblique wave angle $\alpha_b$ and $\theta$ at the wave breaking point (units: degrees).**

| $\alpha_b$ | 2.5 | 5 | 7.5 | 10 | 12.5 | 15 | 17.5 | 20 | 22.5 | 25 | 27.5 | 30 |
|---|---|---|---|---|---|---|---|---|---|---|---|---|
| $\theta$ | 30.7 | 39.8 | 46.7 | 52.5 | 57.7 | 62.5 | 66.9 | 71.2 | 75.2 | 79.1 | 82.8 | 86.5 |





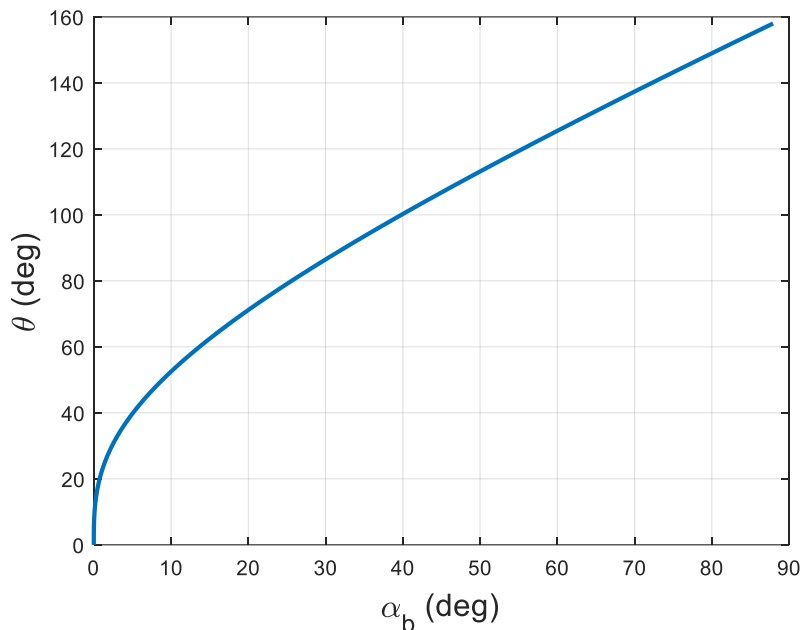

Figure 10: Relationship between oblique wave angle $\alpha_b$ and $\theta$ at wave breaking.

## 4. Analysis of shoreline change caused by oblique waves

In this section, we consider a simple littoral cell, in which the shoreline and depth contour are initially straight and parallel, and the LST is blocked within the groynes. In the first case, we ignored the effect of diffracted waves because of the presence of a short groyne on the left side in Fig. 11. However, with groynes of moderate length and no LST from the updrift coast, wave diffraction occurs, causing erosion in the lee of the shadow zone with a crenulated bay shape because of oblique waves. As the wave action continues, the transition point (for example, C.P. marked as A, B, and C in Fig. 11) at which no erosion occurs on the original shoreline would shift downward with time, while coastal erosion to its left widens. Also in Fig. 11, $t_{1/6}$, $t_{1/3}$ and $t_{1/2}$ denote the times when the control point reaches the points that represent 1/6, 1/3, and 1/2 of the beach length L between the groynes, respectively. At $t \geq t_{1/2}$, the planform remains in equilibrium.



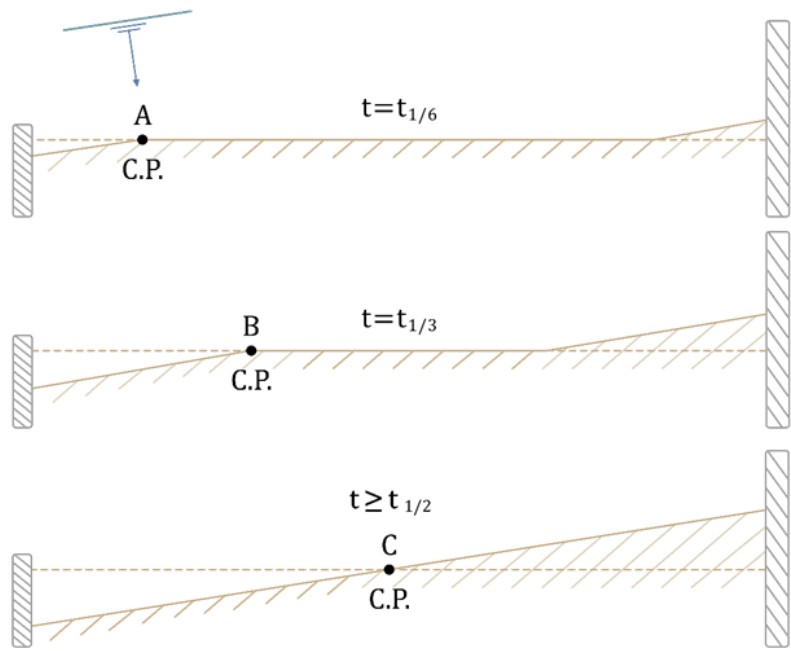

**Figure 11: Temporal variations of downdrift control point without the effect of wave diffraction because of oblique waves within a littoral cell.**

In the case of a short groyne without bay shape formation, the position of the control point $x_c$ can be expressed as a function of the elapsed time t and the LST rate $Q_y$.

$$x_c = \sqrt{\frac{2Q_y t}{(h_c + h_B)\, \tan \alpha_b}} = \sqrt{\frac{2q_b \sin 2\alpha_b t}{\tan \alpha_b}} = 2\cos\alpha_b \sqrt{q_b t} = 2\eta \cos \alpha_b \qquad (10)$$

Here, the LST rate $Q_y$ (unit: m3) is provided by $q_b \sin 2\alpha_b \times (h_c + h_B)$, where $\alpha_b$ is the wave angle at breaking, $q_b$ has units of m2, and $\eta\ (=\sqrt{q_b t})$ is in units of $m\sqrt{t}$. Hence, the position of the control point is a function of the length and time $(\sqrt{t})$, and Eq. (10) can be rearranged for the time elapsed for the control point to reach a distance of $x_c$ such that

$$t = \left(\frac{x_c}{2\sqrt{q_b}\cos\alpha_b}\right)^2 = \left(\frac{1}{2\cos\alpha_b}\right)^2 \tau \qquad (11)$$

Here, $\tau$ is defined as $x_c^2/q_b$. Additionally, the time to reach static equilibrium, $t_{1/2}$, when $x_c = L/2$, is provided by

$$t_{1/2} = \left(\frac{L}{4\sqrt{q_b}\cos\alpha_b}\right)^2 \qquad (12)$$

Equation (12) implies that the time required to reach equilibrium increases as the wave power decreases and the wave oblique angle increases. This approach can be applied to a single littoral cell system affected by wave diffraction around a moderate





to long groyne with protruding length $y_g$, which is located at $\theta = \pi/2$ by using the parallel shoreline approximation of Hsu and Evans (1989) in Eq. (8). Hence, the shoreline advance width, $y_{\pi/2}$, of the parallel shoreline is expressed as

$$\frac{y_{\pi/2}}{y_g} \cong 1 - \frac{2}{\pi}\frac{\beta}{\sin\beta} \tag{13}$$

where it is assumed that $\beta$ converges to zero, $C_0$ and $C_2$ are zero, and $C_1$ is unity.

The cross mark $\times$ in Fig. 12 indicates the maximum indentation position on each sagging shoreline at different time steps. As shown in Fig. 12, we assume the downdrift shoreline orientation to be the same as the wave breaking angle ($\alpha_b$) for applying the PBSE based on the downdrift control point (A, B, and C, respectively) at each time step. Fig. 13 illustrates an example.

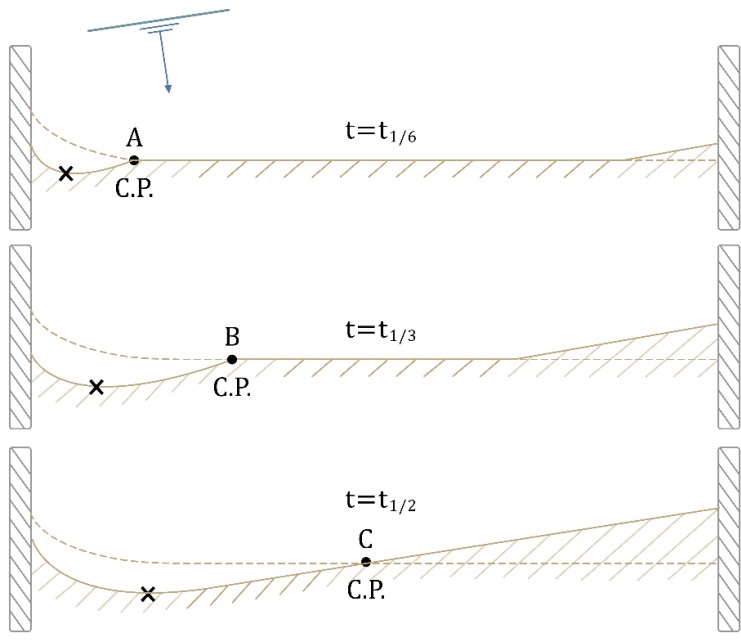

**Figure 12: Temporal change of the control point for apply the PBSE with the effect of wave diffraction because of oblique waves within a littoral cell, also showing maximum indentation at each time step.**



Earth **Surface**
**Dynamics**
Discussions



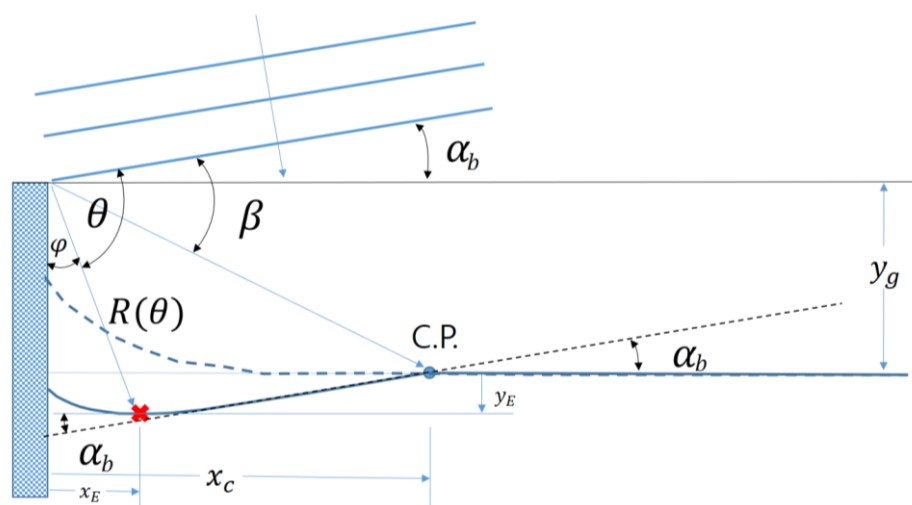

**Figure 13: Application of the PBSE based on the estimation of $\alpha_b$ and downdrift control point (C.P.) and wave diffraction in the lee**
**of a groyne.**

The location of the maximum indentation point $(x_E, y_E)$ shown in Fig. 13 can be determined by using the PBSE approximation
given by

$$x_E = R(\theta)sin\left(\frac{\pi}{2} + \alpha_b - \theta\right) \tag{14a}$$

$$y_E = y_g - R(\theta)cos\left(\frac{\pi}{2} + \alpha_b - \theta\right) \tag{14b}$$

where angle $\theta$ for locating the maximum indentation can be obtained from Eq. (9) for the wave direction $\alpha_b$ at wave breaking.
In addition, $x_E$ is the distance measured from the groyne in the direction of the initial (mean) shoreline, and $R(\theta)$ is a time-
variant function of $x_c$, which can be obtained by Eq. (15) by using the approximation of the PBSE in Eq. (8):

$$R(\theta) \cong \frac{a}{\theta} = \frac{(y_g + x_c \tan \alpha_b)\cos \alpha_b}{\theta} \tag{15}$$

Applying Eq. (15) to Eqs. (14a) and (14b) results in the following alternative expressions for $x_E$ and $y_E$:

$$x_E = \frac{(y_g\cos\alpha_b + x_c\sin\alpha_b)}{\theta}sin(\varphi) \tag{16a}$$

$$y_E = y_g - \frac{(y_g\cos\alpha_b + x_c\sin\alpha_b)}{\theta}cos(\varphi) \tag{16b}$$

where $\varphi = \frac{\pi}{2} + \alpha_b - \theta$. Consequently, a linear relationship for $x_E$ and $y_E$ can be established as

$$y_E = y_g - cot(\varphi)x_E \tag{17}$$

Moreover, Eqs. (16a), (16b), and (17) can be non-dimensionalized by using $y_g$, rendering





$x_E' = \frac{(cos\alpha_b + \eta' sin 2\alpha_b)}{\theta} sin(\varphi)$ (18a)

$y_E' = 1 - \frac{(cos\,\alpha_b + \eta'\,sin\,2\alpha_b)}{\theta} cos(\varphi)$ (18b)

$y_E' = 1 - cot(\varphi)x_E'$ (19)

where $\eta' = \eta/y_g$. Fig. 14 shows the locations of $x_E' = x_E/y_g$ and $y_E' = y_E/y_g$ as a function of dimensionless $\eta' = \eta/y_g$ (0 to

4 with increments of 0.5) for $\alpha_b$ from 1° to 25° with increments of 1°. Fig. 14 indicates that the erosion width $y_E$ increases

with an increase in several parameters (e.g., the protruding length of the groyne $y_g$, $q_b$, $t$, and $\alpha_b$).

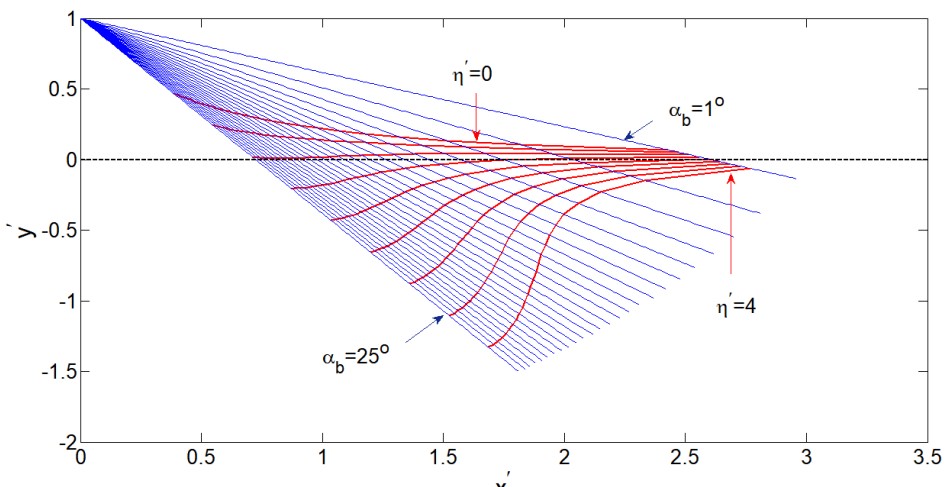

Figure 14: Location of $x_E' = x_E/y_g$ and $y_E' = y_E/y_g$ based on the dimensionless variable $\eta' = \eta/y_g$.

## 5. Results of comparison with numerical model

### 5.1 Shoreline change model

The governing equation for the shoreline change model (Pelnard-Considere, 1957) is a mass conservation equation for

sediment transport. It determines the shoreline change due to the difference in the LST along the coast within the active zone

between the berm and the depth of closure.

$\frac{\partial x}{\partial t} + \frac{1}{(h_c + h_B)}\left(\frac{\partial Q_y}{\partial y} - q\right) = 0$ (20)

where (x, y) are the Cartesian coordinates with x-axis positive pointing seaward, y-axis alongshore, and origin at the MSL,

while $h_c$ and $h_B$ denote the berm height and closure depth, respectively. $Q_y$ is the LST calculated by using the CERC formula

(USACE, 1984), and $q$ represents the cross-shore sediment transport per unit width of the shoreline (Lee and Hsu, 2017).

Within the region of wave diffraction or the vicinity of a coastal structure, an alternative expression is applied to the LST.





$$Q_y = CH_b^{5/2} sin2\alpha_m \tag{21}$$

where $\alpha_m$ is the wave angle within the diffraction zone, which can be determined by using the PBSE (Lim et al., 2021). In the

numerical calculations, we calculated or assigned the quantity of the LST at each grid. For example, we used $Q_y = 0$ for the eroding shoreline along the boundary of the groyne.

## 5.2 Comparison between theoretical and numerical results

Fig. 15 compares the results of theoretical and numerical results for $\tau$ ($=x_c^2/q_b$; units in meters and hours) for three different values of breaking wave direction $\alpha_b$ (10°, 15°, and 20°) at the Jeongdongjin Beach with a natural rocky groyne approximately

80 m long.

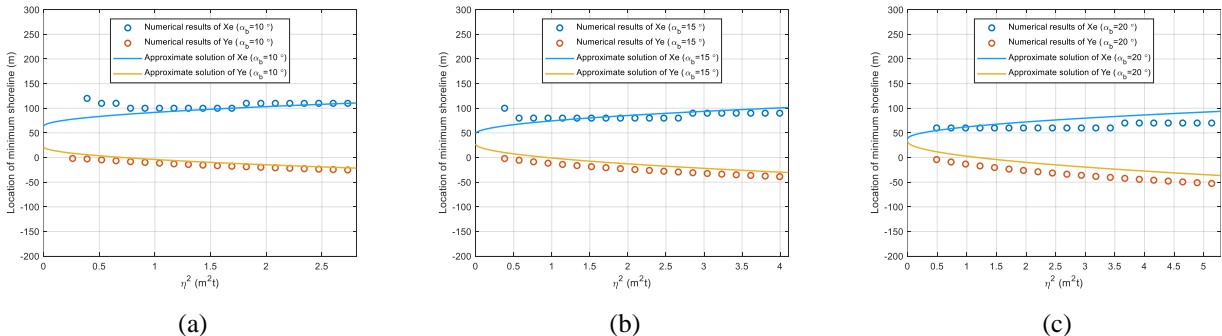

(a)               (b)               (c)

**Figure 15: Comparison between theoretical and numerical results for Jeongdongjin beach with a groyne 80 m long for breaking wave direction of (a) 10°, (b) 15°, and (c) 20°, respectively.**

Taking the most appropriate wave angle of $\alpha_b = 10°$ at the downdrift of the sagging section, we ran the numerical model for a range of time durations (from 6 h to 4 weeks) by using the prototype data for Jeongdongjin Beach—natural groyne with

protruding length $y_g$ = 80 m, beach length L = 850 m, winter high waves of 2.11 m, and LST coefficient C = 0.0847 in Eq. (4). We found the results to be in good agreement with the field observations, as shown in Fig. 16, which reveals that the shoreline adjacent to the groyne advances seaward by the 6th hour before being eroded (landward) afterward.

For the condition of $\alpha_b = 10°$, the mean value of $H_b^{5/2} = 6.5$ and the maximum erosion width of $x_c = 425$ m (i.e., not exceeding L/2), Eq. (16b) gives $x_c = -32.5$ m, which is in good agreement with the numerical estimation result.





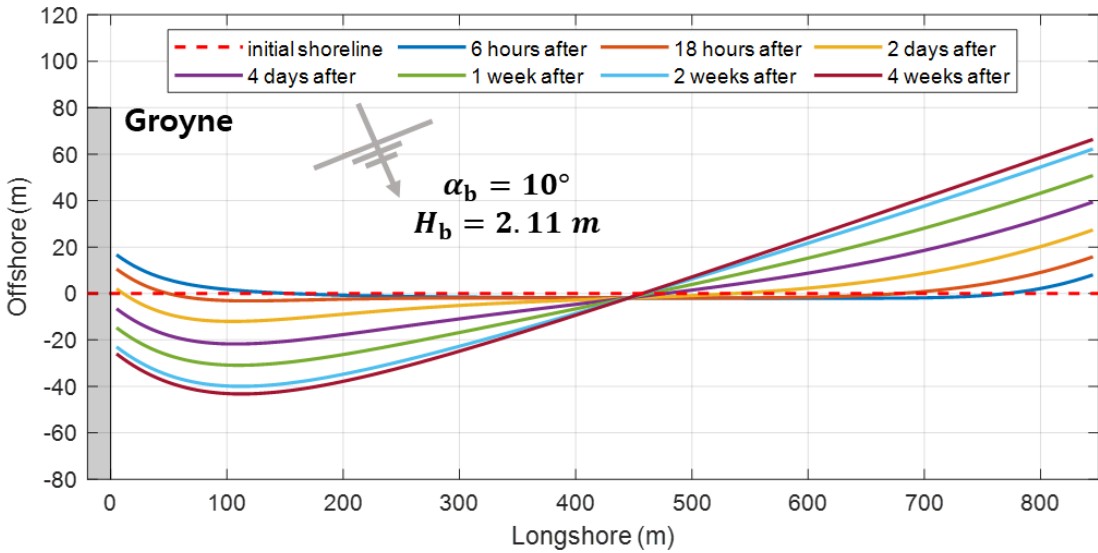


**Figure 16: Results of numerical model, showing the location of downdrift control point for the sagging curve by the groyne and the maximum indentation caused by seasonal high water.**

### 6. Results of comparison with Jeongdongjin monitoring data

As shown in Fig. 17, wave data observed near Jeongdongjin Beach were analyzed to examine whether it is appropriate to apply

the PBSE to this coastal waters. Wave data were observed with an AWAC wave meter at a depth of 32.4 m for 3 years from

September 27th, 2013 to November 21st, 2016. And the distribution of the annual mean wave direction obtained from the data

is shown in Fig. 18. As shown in Fig. 18, the average shoreline formed by the change of the direction of the breaking wave

under the influence of the breakwater after the construction of the gamma-type breakwater and the result of equilibrium

shoreline using PBSE were compared (Lim et al., 2019). Although a simple diffraction wave model was applied, it was

concluded that the application was appropriate to the characteristics of the incident wave observed in the central eastern coast

of Korea by obtaining fairly similar results.



Earth **Surface**
**Dynamics**
Discussions



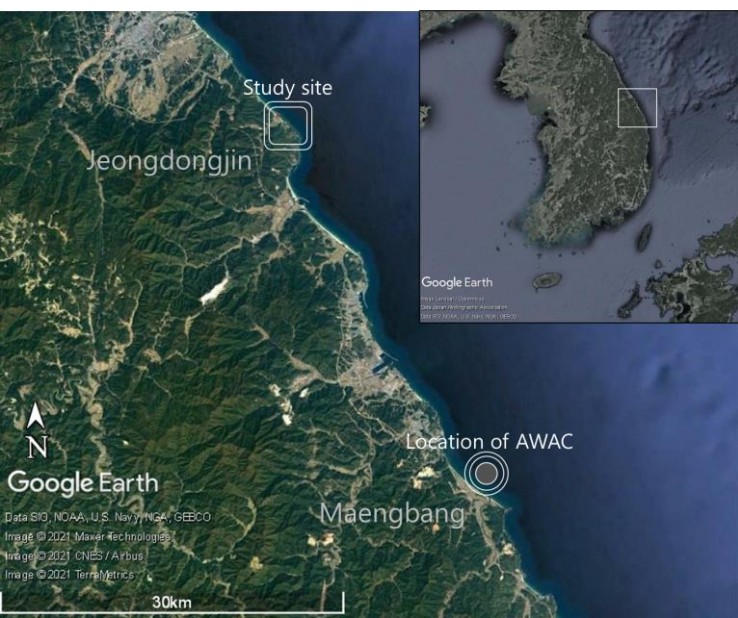

**Figure 17: Location of Jeongdongjin Beach and AWAC wave meter © Google Earth 2021.**

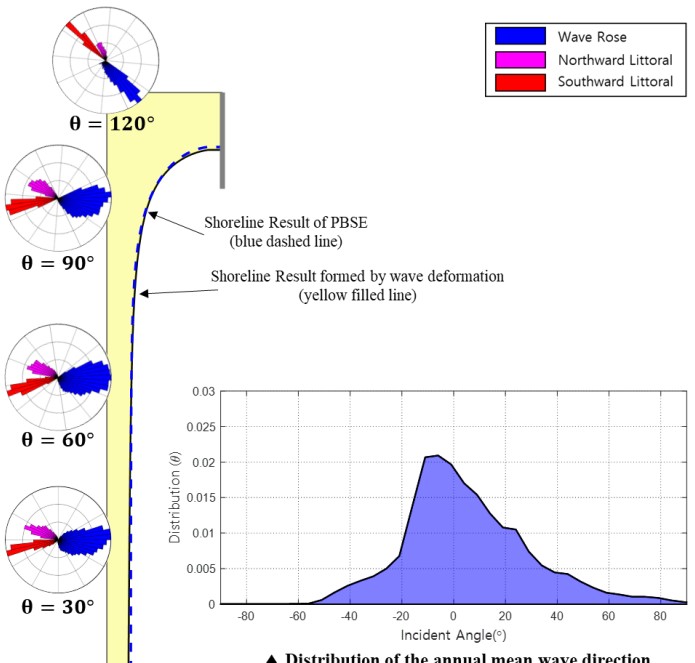

**Figure 18: Comparison of equilibrium shoreline: the average shoreline induced by wave diffraction (yellow filled line) and parabolic bay shape equation (blue dashed line).**

Shoreline monitoring in Korea has been conducted since 2003 as part of the National Coastal Erosion Survey Project. The survey was also conducted to promote efficient coastal maintenance projects in the country via proactive responses based on





scientific data accumulation and analysis; at Jeongdongjin, a video monitoring program that used four cameras commenced in
February 2014, covering 3,280 m (97.3 %) of the local shoreline within a total of 3,370 m (Fig. 19).

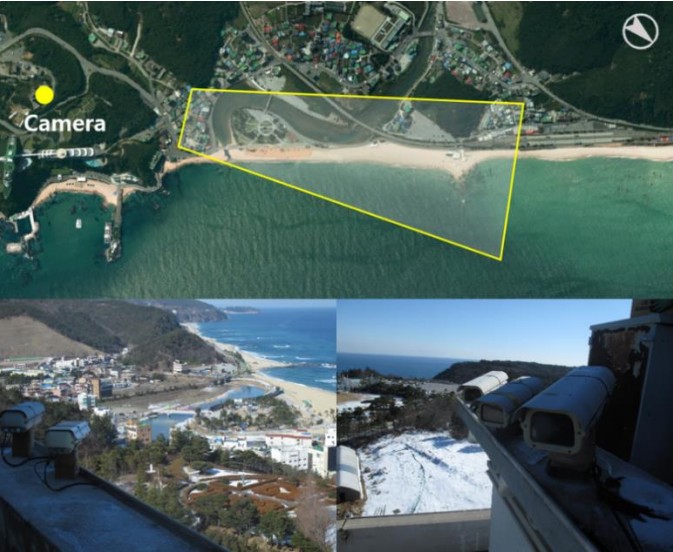

**Figure 19: Video monitoring area covered at Jeongdongjin.**

In this study, the continuously changing shoreline caused by the seasonal waves enables overlapping and averaging of the
pixel values from the 180 instant images taken at 1-s intervals for 3 min. To extract the average pixel value from the images,
we divided the cumulative sum of the attribute values of every pixel by the number of captured images, from which we
determined the coordinates of the ground control points and changing shoreline. To rectify the plane coordinate system for the
shoreline image, we applied the geometric transformation equation of Lippmann and Holman (1989), which transforms the
image coordinates to ground coordinates as follows:

$$y_c^i = f_c tan \left[ tan^{-1} \left( \frac{Y}{Z_c} \right) - \tau \right] \tag{22a}$$

$$x_c^i = \left( \frac{y^2 + f_c^2}{Z_c^2 + Y^2} \right)^{1/2} X \tag{22b}$$

where $x_c^i$ and $y_c^i$ are the Cartesian coordinates in the photographic images; $X$, $Y$, and $Z_c$ are the coordinates of the actual ground
control point position corresponding to the $x$ and $y$ of the photographic image; $f$ is the focal length of the camera; and $\tau$ is the
tilt of the camera. By using this method, we analyzed the images of critical points taken twice a day from December 6–30,
2015, at Jeongdongjin Beach, as shown in Fig. 20, and compared them with the theoretical solution. Nevertheless, note that
the location of the critical points on these images might not include that of the maximum indentation. Therefore, the actual
extent of shoreline retreat may be larger than that presented.



Earth **Surface**
**Dynamics**
Discussions



As shown in Fig. 20, our results of the video monitoring data agree well with those of the theoretical solution for the critical points (i.e., maximum indentation) that used the PBSE approximation, the LST coefficient C = 0.0847, and $(h_c + h_B) = 8$ m for the Jeongdongjin Beach on Korea's eastern coast. In Fig. 20, video monitoring data indicate the location of the critical

points caused by seasonal oblique high waves in November and December 2015, while we obtained the theoretical results from the analysis of the NOAA wave data within the same period of time. Further analysis using the NOAA wave data over 40 years, as shown in Fig. 7(a), indicates that the occurrence of $\alpha_b$ values varying from 0° to 30° may last between zero and 4 months, as shown in Fig. 21, and their corresponding maximum indentations are marked in Fig. 20 for comparison.

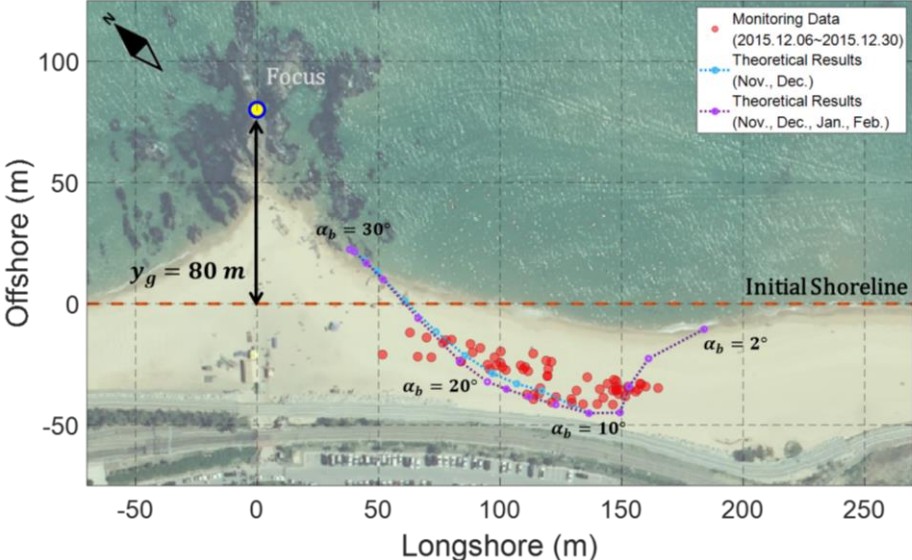

**Figure 20: Comparison between video monitoring data and theoretical solution © National Geographic Information Institute, Korea.**

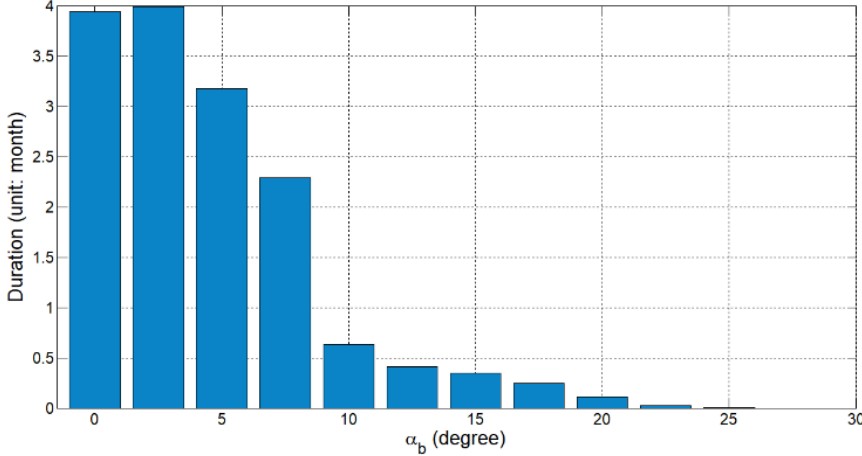

**Figure 21: Duration of each oblique wave angle estimated from NOAA data (unit: month).**





Because our results exclude shoreline retreat because of cross-shore sediment transport, the theoretical solution for the coordinates of the maximum indentation from Eqs. (16a) and (16b) with the LST could be underestimated. In addition, both

theoretical equations neglect sediment bypassing from updrift. Therefore, the solutions must be modified if bypassing occurs. By limiting the alongshore distance of the downdrift control point (at $x_c$ shown in Fig. 13) with the breaking wave angle $\alpha_b$ in relation to the protruding length of the groyne (as shown in Fig. 20), the following relationship can be assumed:

$$x_c^l = \frac{y_g}{tan\alpha_b} \tag{23}$$

Here, the subscript l denotes the limiting value. Table 2 compares the variables $x_c$ and $x_c^l$, obtained from Eqs. (10) and (23),

respectively. If $x_c$ obtained for each $\alpha_b$ is greater than $x_c^l$ obtained for a given $y_g$, $x_c$ in Eqs. (16a) and (16b) should be replaced by $x_c^l$ because bypassing might not occur between 10° and 17.5°. As shown in Fig. 21, the monitoring data in December 2015 support the theoretical solution that the use of $\alpha_b = 10°$ for calculating $x_c$. In Table 2, $\eta\ (=\sqrt{q_b t})$ indicates no effect of protruding length $y_g$ or beach length L (= 850 m), because either the LST is small or the wave duration is too short. However, the erosion width may be reduced as bypassing occurs when the protruding length is short. Thus, the limit in $x_c$ in relative to

L is expected to be within one-half of the beach length, being 425 m (= L/2, where L = 850 m), which is within the range of beach length (approximately 600 m) covered by the video monitoring equipment.

**Table 2: Comparison between the control point locations obtained from NOAA wave data and from Eq. (23).**

| $\alpha_b$ | 2.5 | 5.0 | 7.5 | 10.0 | 12.5 | 15.0 | 17.5 | 20.0 | 22.5 | 25.0 | 27.5 | 30 |
|---|---|---|---|---|---|---|---|---|---|---|---|---|
| $x_c^l$(m) | 1832 | 914 | 608 | 454 | 361 | 299 | 254 | 220 | 193 | 172 | 154 | 139 |
| $x_c$ (m) (Nov., Dec., Jan., Feb.) | 1570 | 1426 | 1209 | 662 | 458 | 374 | 288 | 195 | 92 | 21 | 0 | 0 |
| Constraint* | L | L | L | $y_g$ | $y_g$ | $y_g$ | $y_g$ | $\eta$ | $\eta$ | $\eta$ | $\eta$ | $\eta$ |
| $x_c$ (m) (Nov., Dec.) | 1086 | 993 | 859 | 476 | 325 | 264 | 207 | 137 | 63 | 19 | 0 | 0 |
| Constraint* | L | L | L | $y_g$ | $\eta$ | $\eta$ | $\eta$ | $\eta$ | $\eta$ | $\eta$ | $\eta$ | $\eta$ |

*Parameters in the rows marked Constraint: L for constraint by beach length, η by longshore drift length ($\sqrt{q_b t}$) and $y_g$ by protruding length of groyne.

## 7. Discussion: Engineering countermeasures for mitigating seasonal erosion

Although the characteristics of the seasonal changes in incident waves cannot be modified, the extent of erosion can be reduced either by artificially nourishing the beach to advance the shoreline, or by placing a short groyne to promote sediment accretion within the potentially eroding section. Fig. 22 compares the reduction in potential coastal erosion and its maximum depth $y_E$

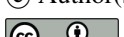



by uniformly nourishing the beach by 10 m and 20 m, respectively, compared with that without artificial nourishment, under the same wave conditions.

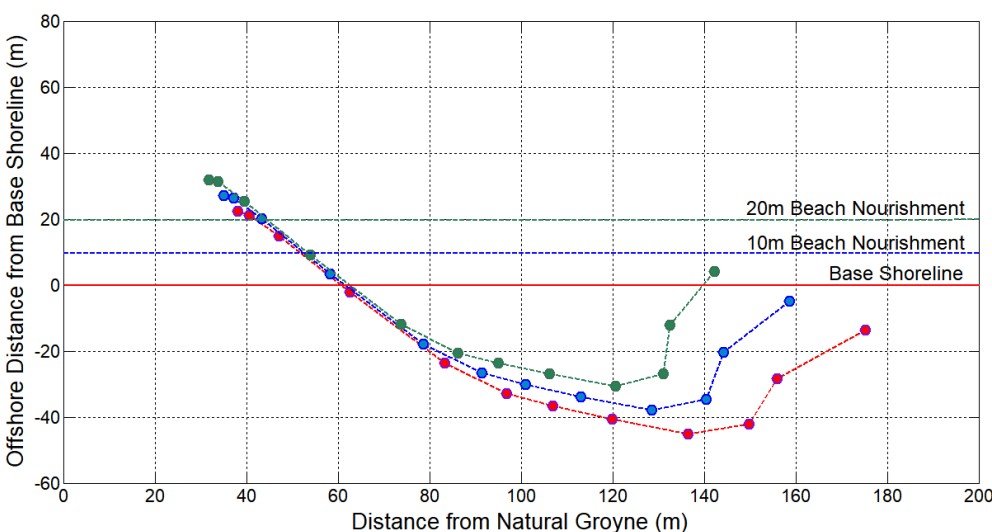

**Figure 22: Location change of maximum erosion width from beach nourishment.**

Fig. 23 shows the temporal shoreline change by placing a short groyne at point G, where it becomes a new downdrift control point for the potential sagging beach curve. Therefore, the dimension of a potential eroding beach at the downdrift of the natural groyne can be reduced by sediment accretion fronting the short groyne. For example, to limit the maximum erosion of $y_E$ within 20 m for $\alpha_b = 10°$ during winter high waves, a groyne may be installed at 327 m from the groyne, for which the location can be estimated by Eq. (16b), leading to

$$x_c = \frac{(y_g - y_E)\theta}{\cos(\varphi)\,\sin\alpha_b} - y_g \cot\alpha_b = \frac{(80+20)\times 52.5\pi/180}{\cos(47.5)\,\sin 10} - 80 \times \cot 10 = 327m \tag{24}$$

Here, $y_g = 80$ m and $\varphi = 47.5°$ for $\alpha_b = 10°$.


Earth **Surface**
**Dynamics**
Discussions

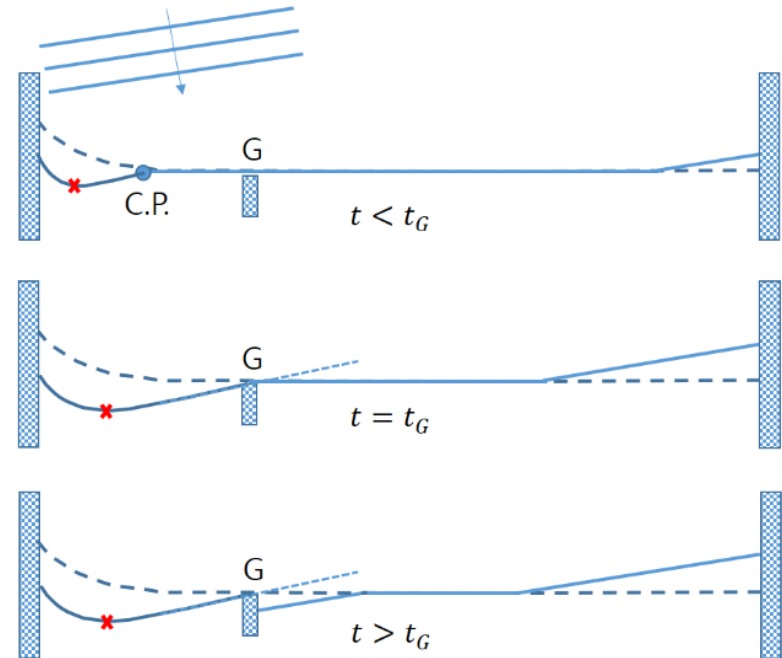

**Figure 23: Controlling maximum erosion width by installing a short groyne.**

## 8. Conclusions

In winter on Korea's eastern coast, severe downdrift erosion often occurs around structures blocking LST because of the inflow of oblique waves. When structures such as groynes and headlands block or delay the LST, erosion damage occurs on the opposite shore. Our study presented a theoretical approach to control downdrift erosion caused by seasonal oblique wave

incidence and to protect the facilities behind it and a solution for coastal engineering. First, by analyzing the 40-year wave data of the NOAA Jeongdongjin coast, we analyzed the wave incidence characteristics by season and theoretically presented the analysis solution of the maximum erosion width by the seasonal oblique wave incidence by analyzing the control characteristics of longshore sediment transport at the study site by using natural rocks. Here, the PBSE proposed by Hsu and Evans (1989) indirectly reflected the diffraction effect of structures with long protrusions. The applicability of PBSE to the Jeongdongjin

coast was verified by comparison with the surrounding equilibrium shoreline and the mean shoreline results obtained by applying the directional wave spectrum to a simple diffraction wave model.

Seasonal erosion on the downdrift side of the structures is mainly related to structural specifications (e.g., protruding length of the structures), wave environment (e.g., duration of wave action as well as wave breaking height and angle), and sand particle size information (e.g., LST coefficient, porosity, and density). The influence varies depending on the location of the control

point, which controls the shape of the shoreline sag. The length unit variables that determine the control point include the beach length, longshore drift length, and protrusion length of the structure. We found the results to be in good agreement with



the yielding line of the maximum erosion width obtained by closed-circuit television (CCTV) in winter in Jeongdongjin. The most critical breaking wave angle of $\alpha_b = 10$ °. was found for a given groyne length of $y_g = 80$ m.

Engineering countermeasures are needed to reduce seasonal hot spots in areas where oblique wave incidence occurs frequently. Reducing the downdrift erosion is impossible because of the seasonal change characteristics of the incident wave or the surrounding sand environment as natural factors. Therefore, in this study, we proposed two major methods to mitigate erosion artificially. The first is to reduce erosion by easing the protrusion length of natural rocks by installing a beach nourishment facility near the hot spot to advance the shoreline. The second is to construct a structure such as a small-scale groyne that acts as a control point to prevent the critical point of the sagging shoreline to retreat to the shore. The engineering approach of this study is expected to aid the design of a groyne to reduce the potential coastal erosion on the downdrift coast, as well as the strategy of nourishing an eroding beach where necessary.

**Data availability**

Not applicable.

**Author contributions**

Supervision, J.L.L.; Writing—original draft, C.L.; Writing—review & editing, C.L., S.H. and J.L.L.; Data acquisition, C.L. and H.S; Visualization, C.L. and J.L.L.;  All authors have read and agreed to the published version of the manuscript.

**Competing interests**

The authors declare no conflicts of interest.

**Acknowledgements**

This research is part of a project entitled 'Practical Technologies for Coastal Erosion Control and Countermeasure' supported by the Ministry of Oceans and Fisheries, Korea.

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
