# Peer review of "An analytical model for beach erosion downdrift of groins: Case study of Jeongdongjin Beach, Korea"

_Earth Surface Dynamics, 2021_

## Referee Comment (RC1)

Dear Dr Orencio D Vinent, Associate Editor

After perusing the MS of "ESurf-2021-71-v1", I would like to submit my comments (**A** and **B** below) and recommendation (**C**) to assist you in reaching a decision on this MS.

Kind regards
Referee 1 (November 1, 2021)

**Referee 1 comments on ESurf-2021-71-v1**

**A. General Comments**

**A1**). This referee has perused "ESurf-2021-71-v1".

**A2**). This referee is familiar with the parabolic bay shape model that the authors adopted to derive analytical equations for estimating the location of the downdrift control point and the maximum indentation point within an eroded crenulate shoreline planform resulting from beach flanking at a natural groin. In the MS, the authors also validate their analytical solution with that from one-line numerical model and shoreline video monitoring data.

**A3**). Like no others (e.g., Balaji et al., 2017; etc.) that apply merely the end results of $R/R_o$ (i.e., relative radius distance from the updrift control point to a point on the static planform) from the parabolic model, these authors have extended the applicability of the parabolic model by deriving new mathematical equations to analyze the spatial and temporal changes of the eroded bay shape. These equations are unique and useful for practical engineers to examining beach flanking shape — a phenomenon that has rarely been taught in the classrooms nor well documented in the literature, despite the erosion at downdrift end of seawalls and groins are common in field condition.

**(A4).** Application of their mathematical equation is straight forward, upon the input of long-term wave data (e.g., breaking wave height and angle, plus longshore sediment transport rate calculated from the input wave condition). This analytical approach will also benefit the readers of Earth Surface Dynamics and other coastal engineering journals on studying beach erosion at the downdrift end of a land-based coastal structure (e.g., seawall or groin).

**A5).** However, considering many research papers on beach erosion/change (e.g., using theoretical, analytical, experimental and numerical) have been published in learned journals and conference proceedings, the authors must state (in 1. Introduction) why they adopt an analytical approach based on the parabolic model for headland-bay beaches, rather by applying some of the existing methods available.

**A6).** The quality of this paper has to be improved first, before the method described in the MS could become available to the readers, by overcoming the shortcomings in the current version of the MS. These include redundant words/phrases, readability and grammar in English, title of the paper, overall re-organization/revision of the contents, sections and sub-sections, as well as limitation of the proposed approach.

**B. Specific Comments**

**B1**). The current title only partially reflects the contents of the paper, and analysis, results and comparison are loosely spread in various sections and sub-sections. Integration should be considered.

For example, please consider revising the title and sections headings, from

Original title: "The sagging shape of shoreline formed on downdrift side of the structures due to seasonal oblique wave incidence"

1. Introduction
2. Analysis of seasonal incident waves
   2.1 Analysis of NOAA data
   2.2 Analysis of seasonal longshore sediment transport
3. Parabolic bay shape equation
4. Analysis of shoreline change caused by oblique waves
5. Results of comparison with numerical model
   5.1 Shoreline change model
   5.2 Comparison between theoretical and numerical results
6. Results of comparison with Jeongdongjin monitoring data
7. Discussions: engineering countermeasures for mitigating seasonal erosion
8. Conclusions

To become:

Revised title: "Analytical approach for beach flanking downdrift of natural groin induced by winter high waves: Case study of Jeongdongj in Korea"

1. Introduction
2. Analytical approach
   2.1 Parabolic bay shape model
   2.2 Downdrift control point
   2.3 Maximum indentation point
3. NOAA's wave data for Jeongdongjin Beach
   3.1 NOAA's wave data (1979 – 2018)
   3.2 Seasonal longshore sediment transport
4. Results and comparison
   4.1 Analytical results versus numerical results
   4.2 Analytical results versus shoreline monitoring results
5. Discussions
6. Concluding Remarks

**B2**). Abstract [L9–21]:    Note: "L" for Line number in the original MS.

Text written loosely. Please rewrite. Please also provide keywords.

**B3**). 1. Introduction [L23–78]

Pleas define "beach flanking" at the beginning of the Introduction and describe the effect of groin on sediment transport.

Please describe methods and/or models (numerical etc.) for analyzing beach erosion, and show references.

Please replace "sagging" by an appropriate word throughout.

[L23] "… the eastern coast of …" → "… the east coast of …"?

[L31] "… a group of natural (pillar) rocks…" → "…a cluster of natural pillar rocks…"

[L36] "Many studies….". Please provide references.

[L42] "the LST becomes static to maintain the shoreline planform…". Is "becomes static" correct?

[L44] "…for project planning." → "…for project planning (USACE, 2002)"

[L47] "To reproduce the shapes of shorelines in the laboratory," → please revise.

[L54] "…sagging shape…" →"…crenulate shape…"

[L56] "High resolution numerical models…". High resolution?

[L58–59] "These laboratory experiments … contributed to solving scientific equations about erosion by providing a similar sagging shape." What do you want to say? Please revise.

[L60] "… still lack reliable factor analysis." What is 'factor analysis'? Is this a proper technical term?

[L61] "This study developed a theoretical approach, …". Past tense? 'develop'?

[L61–64] Please use verbs in present tense.

[L64–65] "…formed frequently…"?,   "…a group of…"?

[L61–78] "theoretical approach", "theoretical analysis" and "theoretical solutions" etc. → Use "analytical approach", "analysis" and "analytical solutions"?

Please avoid repeating the use of some expressions, such as "Section 2…." at the beginning of a sentence or "…in Section X." at the end of a sentence.

[L77] "… through a theoretical formula that facilitates factor analysis."?? Please revise/clarify.

**B4**). 2. Analysis of seasonal incidence waves [L79] → please change section heading.

2.1 Analysis of NOAA data [80–99] >>> Relocate this section to new Sect. 3.1?

[L86; L97; L99] "eastern coast" → "east coast"?

[L88] "…cause by the energy of the oblique high waves."? Please revise.

[L95–96] "…Average over 10 year-intervals..." → "…Averaged over every 10-year period…"?

[L97–98] "…local shoreline orientation in Gangwon-do was approximately N43ºE." → "…local shoreline orientation at Jeongdongjin is in WN to SE direction, or N133ºE."?

**B5).** 2.2 Analysis of seasonal longshore sediment transport [L100–160] >>> Relocate entire section to new Sect. 3.2?

[L101–146] All "$Q_y$", "$P_y$", "$I_y$" → "$Q_l$", "$P_l$", "$I_l$"? because subscript "y" is not defined, while subscript "$l$" is the most used term in textbook.

All the "sin" and "cos" should not be italicized.

[L118] "…the isobath of the seabed…" → "… seabed contours…"?

[L121] How to derive Eq. (5), reference?

[L133–138] Paragraph and the original Fig. 5 may be deleted?

[L151–152] "…high waves in winter… arrives from…N38ºE-2.5º to N38ºE +7.5º" >> but $\alpha_b$ = 10º is used in Figs. 16 and 20, Why?

**B6).** 3. Parabolic bay shape equation [L164–180] >>> Renumbered as new Sect. 2.1?

[L165] "… is provided by…" → "…defines the location of a point $P$ ($R$, $\theta$) on an embayed beach in static equilibrium by…"

[L166–180] All the "sin" and "cos" should not be italicized.

[L169] "…between the wave crest line (wave crest base line) and the line that passes through the …parallel to the shore base line" → "… between the wave crest base line (at the focus) and the tangential line at …on the shore base line;"

[L173] "…= 0 is satisfied to endure…" → "…= 1.0 (unity) to ensure…"

[L180] Eq. (9). Please show key interim steps that lead to this equation.

**B7).** 4. Analysis of shoreline change caused by oblique waves [L184–242] >>> Please re-number the section and change the heading as new Sect. 2.1 and Sect. 2.2

[L199] Pease show interim steps leading to Eq. (10).

[L199–237] All the "sin", "cos" and "cot" should not be italicized.

[L200–201] Unit "m3" and "m2" → superscript "m$^3$" and "m$^2$"

[L204] "Additionally, the time to reach static equilibrium, $t_{1/2}$, when $x_c$ = L/2, is provided by" → "When static equilibrium is reached for $x_c$ = L/2, the time".

[L206] "… wave power decreases…"??

[L210] Please show key interim steps leading to Eq. (13).

[L223–224] Please show key interim steps leading to Eqs. (14a) and (14b).

[L241–242] Please consider to replot Fig. 14 using "$\alpha_b$" values of 0º, 2.5º, 5º, 7.5º, 10º, 15º and 20º, for better viewing and manual application.

**B8**). 5. Results of comparison with numerical model [L243–272] >>> Re-number section as new Sect. 4 and also change section heading as: Results and comparison

5.1 Shoreline change model [L244–256] >>> First part in new Sect. 4.1: Analytical results with numerical results

[L245–247] "…for the shoreline change model…sediment transport. It determines the shoreline change due to the difference in the LST along the coast within the active zone between the…" → "… for shoreline change…within a control volume due to the difference in LST across the active beach zone from berm…"

[L248 Eq. 20; L250; L253 Eq. 21; L255] → Replace all four "$Q_y$" by "$Q_l$", to be consistent with the LST in the existing Sect. 2.2 (or new Sect. 3.2)?

[L255–256] "…, we calculated or assigned the quantity of the LST at each grid. For example, we used $Q_y = 0$ for the eroding shoreline along the boundary of the groyne." → "…, the quantity of LST at each grid is calculated or assigned. For example, $Q_l = 0$ is assigned along the updrift groin where shoreline is receding."

**B9**). 5.2 Comparison between theoretical and numerical results [L257–272] >>> Second part in new Sect. 4.1: Analytical results with numerical results

[L269] "…gives $x_c = -32.5$ m, …" → "… gives $y_E = -32.5$ m, …"?

[L270–272, Fig. 16] → Replace the "Offshore (m)" on the ordinate by "Cross-shore (m)"

**B10**). 6. Results of comparison with Jeongdongjin monitoring data [L 273–317] >>> Renumber Sect. and heading as in new Sect. 4.2: Analytical results versus shoreline monitoring results

[L274–276] "As shown in Fig. 17, where … to September 27th, 2013 to November 21st, 2016." >>> to be relocated after the new Fig. 10 at the end of new Sect. 3.1, and the two original sentences are revised as "Nearshore wave data were collected by an AWAC wave meter at a depth of 32.4 m to the south of Jeongdongjin Beach. From the data recorded over three years (27 September 2013 to 21 November 2016), the distribution of the annual mean wave direction is plotted (Fig. 11). As shown in Fig. 11, the prevailing wave direction was within -15º and +10º from the normal to the shoreline."

[L276–281] "As the distribution of the annual …obtaining fairly similar results." >> Suggestion: Please delete this part, as it may be irrelevant.

[L282–283] Original Fig. 17 to be re-numbered as new Fig. 9 and relocated under new Sect. 3.1.

[L283–284] Original Fig. 18: The shoreline with gama-groin and legend may be

deleted and just retain the little plot (called new Fig. 11) that shows the 'Distribution of the annual mean wave direction'.

[L285–286] Figure caption for the original Fig. 18 may be revised as:

"Fig. 11: Distribution of annual mean wave direction obtained from AWAC meter during 2013 – 2016."

[L287–290] "Shoreline monitoring in Korea…Project. The survey as also conducted to promote…based on scientific data accumulation and analysis; at Jeongdongjin, a video monitoring program that used four cameras…, covering 3,280 m (97.3%) of the total shoreline within a total of 3,370 m (Fig. 19)" → "Shoreline monitoring in Korea…Project, aiming to promote…, based on data collection and analysis. At Jeongdongjin, a video monitoring program employing four cameras…, which covers 3,280 m (97.3%) of a total of shoreline about 3,370 m (Fig. 18)."

Question: Are the values of "3,280 m" and "3,370 m" correct??

Please double check the correctness of the length of shoreline cited in the original MS [L290], because the length of Jeongdongjin Beach is only about 800 m (see [L330] in the original MS).

[L293] "the continuously changing shoreline…" → "the spatial and temporal change in shoreline…"

[L295–303] Delete four "we"s in "we divided…", "we determined…", "we applied…" and 'we analyzed…"

[L294–296] "images, we divided the accumulative sum…every pixel by the number of captured images, from which we determined the coordinates…and changing shoreline." → "images, the cumulative sum…of every pixel is divided by the number of the images captured, to determine the coordinates… and changing shoreline."

[L297] "image, we applied the geometric transformation equation of Lippmann and Holman (1989), which transforms…" → "image, the geometric transformation equation given by Lippmann and Holman (1989) is applied to transform…"

[L299] The two "*tan*"s in Eq. (22a) should not be italicized.

[L303–304] "By using this method, we analyzed the images of critical points taken twice a day from December 6 – 30, 2015, at Jeongdongjin Beach, as shown in Fig. 20, and compared them with the theoretical solution." → "Images of the critical points that were taken twice daily on Jeongdongjin Beach during December 6 – 30, 2015 are analyzed and compared with the analytical solution, as marked in Fig. 20."

[L307–308] "…, our results of the video…with those of the theoretical solution for

the…) that used the PBSE approximation,…" → "…, the results of video…with that of the analytical solution for the…) predicted by the analytical approach, …"

[L309–311] "…video monitoring data… in  December 2015, while we obtained the theoretical results from the analysis of the NOAA wave data within the same period of time." → "…video monitoring data … in December 2015, while the analytical results are calculated applying the NOAA wave data within the same period."

**B11**). 6. Results of comparison with Jeongdongjin monitoring data [L318–333]

[L318–333] Please relocate this part to new Sect. 5: Discussions (1)

[L318] "Because our results exclude shoreline retreat because of cross-shore sediment transport, the theoretical solution…" → "Due to cross-shore sediment transport is excluded in the present study, the analytical solution…"

[L319–320] "…, both theoretical equations neglect…." → "…, the mathematical equations (i.e., Eqs. (22a) –(22b)) neglect…"

[L323] Eq. (23): "tan" should not be italicized.

[L324–325] "Here, the subscript 1 denotes the limiting value. Table 2 compares the variables $x_c$ and $x_c^l$, obtained from …, respectively. If $x_c$ obtained for each $\alpha_b$ is greater than $x_c^l$ obtained a given $y_G$,"

→ "where subscript $l$ denotes the limiting value. Variables $x_c$ and $x_c^l$, obtained from …, respectively, are compared in Table 2. If $x_c$ for each $\alpha_b$ is greater than $x_c^l$ for a given $y_G$,"

[L327] "…the theoretical solution that the use of $\alpha_b = 10^o$ for…" → "…the analytical solution that uses $\alpha_b = 10^o$ for…"

**B12**). 7. Discussions [L334–350] >>> Renumber this section as new "5. Discussions"
[L334–350] Please relocate this part to new Sect. 5: Discussions (2)

- Please expand the Discussion section by including description on the limitations of the analytical approach presented in this study, as compared with other known theoretical methods and/or numerical models for predicting shoreline changes!

**B13**). 8. Conclusions [L351–376] >>> Renumber as new "6. Concluding remarks"
[L351–376] Please revise.

**B14**). References [L387–444]
Please double check the references, and remove redundant list.
>>> Please add the following references:

Bakker, W.T., 19??. The influence of longshore variation of the wave height on the littoral current. Report WWK71-19, Ministry of Public Works, The Netherlands.

Balaji, R., Kumar, S.S., Misra, A., 2017. Understanding the effects of seawall construction using a combination of analytical modelling and remote sensing techniques: Case study of Fansa, Gujarat, India. J. Ocean and Climate Systems 8 (3), 153–160.

Ozasa, H., Brampton, A.H., 1980. Mathematical modeling of beaches backed by seawalls. Coastal Eng. 4 (1), 47 –64.

**C. Recommendation**

**C1**). Major revision is required to improve the quality of this manuscript, prior to resubmittal.

***** END of REPORT ****

---

## Referee Comment (RC2)

Review of **The sagging shape of shoreline formed on downdrift side of the structures due to seasonal oblique wave incidence**
by Lim et al

November 2021

Authors present a method based on the parabolic static equilibrium beach to estimate the main dimensions of the expected shoreline indentation downdrift groins. The method is then compared against numerical model results and applied to a beach in Korea. The manuscript addresses a topic covered by Earth Surface Dynamics and, in this sense, the manuscript can be of interest for many ESurfD readers.

In what follows, some observations/comments/suggestions are given.

**General comments**

**[1]** The manuscript needs a thorough revision of the **English language**. Grammatical errors (and unusual sentence constructions) are very frequent throughout the manuscript and will not be indicated here except in selected cases.

**[2]** The **title** is long and confusing. Please make it simpler. "Sagging shape" is not usually employed in the context of the topic, it is usually referred as shoreline indentation or erosion.

**[3]** Throughout the manuscript you mix data, methods and results in the same sections. This is a bit confusing for readers. Please re-organise the manuscript to include the following sections and restrict the content to the parts corresponding to the heading:

(i) **Study area and data** where you describe the study site characteristics and present *all data* to be used;

(ii) **Methodology** where you present and describe *all methodology* used in the study;

(iii) **Results** where *all results* are presented;

(iv) **Discussion**, where you discuss the obtained results and the applicability of the presented approach/model; and

(v) **Conclusions.**

**[4] Description of methods**. At its present version, some of the text included in the description of the methods is "excessive". For instance, most of text describing the longshore sediment transport formula can be just simplified in a single line stating which is the formula to be used plus the formula itself. Most of this text can be found in any textbook. Please revise and simplify when possible.

In some cases, you provide more information than strictly needed. For instance, in section 6 the text from line 294 to 303 can be summarise in something like "shoreline positions were derived from time-averaged video images by using the method proposed by X". Equation 22 is not needed.

**[5] Introduction**

In this section you mix a series of concepts related to shoreline development. However, it is not clear why you do it. It would be better for readers if you clearly motivate your study.

**[Lines 61-65]** Please, reformulate the paragraph and include a sentence where you explicitly state the objective of the paper. Something like "The main aim of this work is ……."

**[lines 147 – 152 & Fig 7]** How did you calculate $H_b$? How did you obtain $\alpha_b$?

**[lines 204-205]** Define L (groin spacing / bay length)

**[line 208]** *parallel shoreline approximation* of Hsu & Evans?

**[lines 252-253]** *If you are going to propose an alternative expression for LST in the diffraction zone you have to consider not only changes in the wave angle, as you do in equation (21), but you should also include a term to account for currents induced by the wave height gradient occurring in the diffraction zone (e.g. Osaza & Brampton'1980 approach).*

**[line 263]** why is $\alpha_b$=10 the most appropriate angle?

**[lines 266-267]** Fig 16 does not indicate any agreement with field observations. Fig 16 just shows shoreline evolution as predicted by the model.

**[line 269]** I think that the xc = -32.5 is wrong, probably you refer to another thing and not to xc.

**[line 274]** Fig 17 is only a map where the location of AWAC is shown. It does not provide any crucial information (it should be enough to say that wave data were recorded by an AWAC system southward of the study area).

**[lines 277-281]** Can you explain better what did you do? What do you refer with average shoreline? Which diffraction model?

**[lines 307-313]** Here you are comparing nearly instantaneous critical points detected from hourly/daily video observations with predictions done for average seasonal conditions. Is this consistent to say that data agree?

**[line 324]** subscript -> superscript

**Discussion**

Here, you don't formally discuss your work (neither the shortcomings of the methodology or the results obtained). You should include here the advantages and disadvantages of the method. Why am I going to use your approach?

**Conclusions**

This section is too large (it is larger than the *Discussion* section). It is more a summary than conclusions.

**Figures**

**Figure 2.** please mark the location of the study site in the photo of Korea.

**Figure 10** and **table 1** are providing the same info. Also, they have been obtained by applying equation 9 to selected θ values. You could remove them (figure and table) without affecting the understanding of the text. In any case, please remove at least one of them (table or figure).

**Figure 19** is not strictly needed.

---

## Author Comment (AC1)

**Response to Referee reports for ESurf-2021-71**

R#: Referee number (1 or 2); C#: Comment/Response number; A: Authors' response.

Line number in **Original MS** : OL #

Line number in **Marked copy** : ML #

**General Comments from Referee 1 and authors' response**

This paper attempted an analytical approach to beach flanking, which had many cases of damage in the east coast of Korea. As noted in the general comment on Referee 1, this analytical approach will also benefit the readers of Earth Surface Dynamics and other coastal engineering journals on studying beach erosion at the downdrift side of structures blocking longshore sediment.

In order to improve the readability of the research that has not been tried much, the paper was improved overall based on the comments of referee 1. These include redundant words/phrases, readability and grammar in English, title of the paper, overall re-organization/revision of the contents, sections and sub-sections, as well as limitation of the proposed approach. And the corrected and supplemented details are in the following answers of specific comments from referee 1.

**Specific Comments from Referee 1 and authors' response**

**R1C1)** The current title only partially reflects the contents of the paper, and analysis, results and comparison are loosely spread in various sections and sub-sections. Integration should be considered.

**AC1)** The paper has been revised with the title and contents as recommended.

Revised title: "Analytical approach for beach flanking downdrift of natural groin induced by winter high waves: Case study of Jeongdongjn in Korea"

Revised contents:

1. Introduction

2. Analytical approach

    2.1 Parabolic bay shape model

    2.2 Downdrift control point

    2.3 Maximum indentation point

3. NOAA's wave data for Jeongdongjin Beach

    3.1 NOAA's wave data (1979 – 2018)

    3.2 Seasonal longshore sediment transport

4. Results and comparison

    4.1 Analytical results versus numerical results

    4.2 Analytical results versus shoreline monitoring results

5. Discussions

6. Concluding Remarks

**R1C2)** Abstract [OL 9–21]: Text written loosely. Please rewrite. Please also provide keywords.

**AC2)** The abstract of the paper was compactly supplemented, and the contents and limitations of the entire approach were rewritten in the abstract. The amended contents are as follows.

[ML 11–21] Beach erosion at the unprotected downdrift end of a seawall or revetment is common and the same can be found at a groin, when oblique waves approach these structures. This phenomenon, known as beach flanking, has often occurred at downdrift of natural groin on the east coast of South Korea during high waves in winter months. The resulting planform assumes a distinctive crenulate shape with a maximum indentation point within the eroded shoreline. Analytical approach is employed to study the beach erosion at downdrift end of a natural rocky groin on Jeongdongjin Beach in Korea, using mathematical equations derived from the parabolic model for headland-bay beaches in static equilibrium, to predict the downdrift control point and maximum indentation of the eroded shoreline planform. NOAA's wave data over 40 years are used to determine the prevailing wave height, wave angle at breaking, wave direction and longshore sediment transport rate to solve the equations. Location of the calculated maximum indentation is verified using shoreline video monitoring data, and its spatial and temporal change is also compared with the result of one-line numerical model for shoreline change. Limitation of the proposed analytical approach for Jeongdongjin Beach is discussed, as well as the effect of sediment bypassing the groin.

[ML 23] Keywords: Beach flanking, Analytical approach, Parabolic model, Maximum indentation point, Shoreline video monitoring.

**R1C3-1)** 1. Introduction [OL 23–78]

Please define "beach flanking" at the beginning of the Introduction and describe the effect of groin on sediment transport.

Please describe methods and/or models (numerical etc.) for analyzing beach erosion, and show references.

Please replace "sagging" by an appropriate word throughout.

**AC3-1)** First of all, I modified the term "beach flanking" rather than "sagging" as a whole. The term "beach flanking" was cited as a term expressed by Balaji et al. (2017), who attempted a similar study. And at the beginning of the introduction, the terminology "beach flanking" was explained, and methods for analyzing beach erosion (include the effect of groin on sediment transport) was also described. The amended contents are as follows.

[ML 25–38] Beach erosion at the downdrift end of a shore-based coastal structure (e.g., seawall and groin) is common, but rarely being taught in the classrooms nor well documented in the literature. This phenomenon is known as beach flanking, which results in localized eroding beach in crenulate shape. Looking at the history of shore protection, seawalls of vertical or sloping (revetments) have been used for many decades as a purported protection in an erosion situation. It is however unfortunate that they have often promoted further erosion, not only the beach in front of them receding to the point of being non-existence, due to partial standing and reflecting waves, but also short-crested system when waves approach obliquely which accelerates sediment removal to downdrift coast. Consequently, immediately at the unprotected downdrift end of seawalls, erosion occurs, promoting by the action of diffraction-refraction and nearshore currents to remove sediment away and produced a seaward-concave shoreline planform (e.g., Kraus and McDougal, 1996). On the other hand, groins have been installed in England since the 16th century. They are structures of moderate dimension which run from the beach into the sea. The structures are generally at right angles to the shoreline, inclined or in complex form in more recent time. The purpose of installing groins is to intercept and accumulate sediment on the updrift side, during periods of oblique persistent swell. Whilst the sediment being accreted, the beach downdrift of these structures will suffer erosion. Only after the updrift shoreline has built up to the tip (head) of a groin, or nearly so, sediment will again be fed to the downdrift side, to recover the receded shoreline.

**R1C3-2)** [OL 23] "… the eastern coast of …" → "… the east coast of …"?

[OL 31] "… a group of natural (pillar) rocks…" → "…a cluster of natural pillar rocks…"

[OL 44] "…for project planning." → "…for project planning (USACE, 2002)"

[OL 54] "…sagging shape…" → "…crenulate shape…"

**AC3-2)** We modified it to the term recommended.

[ML 39] "… the east coast of …"?

[ML 46] "…a cluster of natural pillar rocks…"

[ML 77] "…for project planning (USACE, 2002)"

[ML 56] "…crenulate shape…"

**R1C3-3)** [OL 36] "Many studies….". Please provide references.

**AC3-3)** References related to the study of the negative effects of beach erosion at downdrift of groins were added.

[ML 48-49] Beach erosion at downdrift of groins (Fig. 2) and their negative impact on downdrift beach has been well understood (Lehnfelt and Svendaen, 1958; Bakker, 1968, Bakker et al., 1970; Price and Tomlinson, 1970; Magoon and Edge, 1978; Headland et al., 2000; USACE, 2002).

**R1C3-4)** [OL 42] "the LST becomes static to maintain the shoreline planform…". Is "becomes static" correct?

**AC3-4)** The content that was intended to be delivered was delivered incorrectly. So we modified it as follow.

[ML 74-75] When this condition is reached, LST is not required to maintain the shoreline stability within an embayment under the same predominant wave direction (Hsu et al., 2000).

**R1C3-5)** [OL 47] "To reproduce the shapes of shorelines in the laboratory," → please revise.

**AC3-5)** The content that was intended to be delivered was delivered incorrectly. So we modified it as follow.

[ML 50-52] Badei (1994) conducted laboratory experiments to reproduce the shoreline planform and topographic changes, while Wang and Kraus (2004) performed tests on erosion without longshore sediment transport (LST).

**R1C3-6)** [OL 56] "High resolution numerical models…". High resolution?

**AC3-6)** The content that was intended to be delivered was delivered incorrectly. So we modified it as follow.

[ML 80-81] … ,supported by numerical models (e.g., Xbeach and Sbeach) that include waves, currents and topographic changes.

**R1C3-7)** [OL 58–59] "These laboratory experiments … contributed to solving scientific equations about erosion by providing a similar sagging shape." What do you want to say? Please revise.

[L60] "… still lack reliable factor analysis." What is 'factor analysis'? Is this a proper technical term?

**AC3-7)** The content that was intended to be delivered was delivered incorrectly. So we deleted it.

**R1C3-8)** [OL 61] "This study developed a theoretical approach, …". Past tense? 'develop'?

[OL 61–64] Please use verbs in present tense.

[OL 64–65] "…formed frequently…"?, "…a group of…"?

[OL 61–78] "theoretical approach", "theoretical analysis" and "theoretical solutions" etc. → Use "analytical approach", "analysis" and "analytical solutions"?

Please avoid repeating the use of some expressions, such as "Section 2…." at the beginning of a sentence or "…in Section X." at the end of a sentence.

[OL 77] "… through a theoretical formula that facilitates factor analysis."?? Please revise/clarify.

**AC3-7)** The parts of the introduction were somewhat ambiguous, so it was supplemented as a whole. The amended contents are as follows.

[ML 82-97] The aims of this paper are threefold, (1) to study the beach flanking at Jeongdongjin Beach in Korea (Figs. 1b and 3), (2) to demonstrate the applicability of an analytical approach derived from the parabolic model, and (3) to develop mathematical equations for calculating the most critical point (i.e., maximum indentation) in the eroded crenulate shape. To solve these equations, NOAA's wave data over 40 years are used to determine the prevailing wave heights and directions in winter months to Jeongdongjin Beach, where a cluster of natural pillar rocks protruding into the sea behaving like a groin. The analytical results calculated are then validated by comparing with the results from the monitoring video images and the numerical model of shoreline change.

In addition to the introduction given in section 1, section 2 in this paper describes the analytical approach using the parabolic bay shape model (Hsu and Evans, 1989) in static equilibrium, from which mathematical equations are derived for the downdrift control point and the maximum indentation of the eroded beach. Analysis of NOAA's wave data over 40 years is presented in section 3.1 to provide averaged wave heights and wave angles at breaking points as input to the mathematical equations derived in sections 2.2 and 2.3, as well as the seasonal longshore sediment transport rate (section 3.2) for one-line shoreline change model (section 4.1). The analytical results calculated for the maximum indentation point are then compared with the results of numerical method applying one-line shoreline change (section 4.1) and that from shoreline video monitoring program for Jeongdongjin Beach (section 4.2). Finally, discussions on the limitation of the proposed analytical approach and effect of sediment bypassing are given in Section 5. Concluding remark is given in Section 6.

**R1C4-1)** 2. Analysis of seasonal incidence waves [OL 79] → please change section heading.

**AC4-1)** The title of the section heading was specifically modified as follow.

Revised contents: "NOAA's Wave Data for Jeongdongjin Beach"

**R1C4-2)** 2.1 Analysis of NOAA data [OL 80–99] >>> Relocate this section to new Sect. 3.1?

**AC4-2)** That section was relocated to new Sect. 3.1.

**R1C4-3)** [OL 86; OL 97; OL 99] "eastern coast" → "east coast"?

**AC4-3)** We modified it to the term recommended.

**R1C4-4)** [OL 88] "…cause by the energy of the oblique high waves."? Please revise.

**AC4-4)** The content that was intended to be delivered was delivered incorrectly. So we modified it as follow.

[ML 254-255] The wave data are analyzed and the results used to calculate the change of the eroding shoreline curve at downdrift of the natural rocky groin caused by the oblique high waves in the winter.

**R1C4-5)** [OL 95–96] "…Average over 10 year-intervals..." → "…Averaged over every 10- year period…"?

[OL 97–98] "…local shoreline orientation in Gangwon-do was approximately N43°E." → "…local shoreline orientation at Jeongdongjin is in WN to SE direction, or N133°E."?

**AC4-3)** We modified it to the term recommended.

[ML 261-262] … averaged over every 10-year intervals …

[ML 263-264] … and the local shoreline aligns in NW – SE direction (about N133°E).

**R1C5-1)** 2.2 Analysis of seasonal longshore sediment transport [OL100–160] >>> Relocate entire section to new Sect. 3.2?

**AC5-1)** That section was relocated to new Sect. 3.2.

**R1C5-2)** [OL 101–146] All "$Q_y$", "$P_y$", "$I_y$" → "$Q_l$", "$P_l$", "$I_l$"? because subscript "y" is not defined, while subscript "l" is the most used term in textbook.

All the "sin" and "cos" should not be italicized.

[L118] "…the isobath of the seabed…" → "… seabed contours…"?

**AC5-2)** We modified it to the term recommended.

**R1C5-2)** [OL 121] How to derive Eq. (5), reference?

**AC5-2)** Eq. (5) is derived using longshore sediment transport equation (see the book of Dean and Dalrymple (2001) : Coastal Processes with Engineering Applications, "Littoral Drfit Compuations based on Deep Water Data" parts) and is not cited separately.

**R1C5-3)** [OL133–138] Paragraph and the original Fig. 5 may be deleted?

**AC5-3)** The content was unnecessary, so we deleted it.

**R1C5-4)** [OL 151–152] "…high waves in winter… arrives from…N38°E-2.5°  to N38°E +7.5°" >> but $\alpha_b$ = 10° is used in Figs. 16 and 20, Why?

**AC5-4)** The results of wave direction was obtained by analyzing NOAA data. However, in the period of severe erosion damage in Jeongdongjin Beach, waves entering with a larger oblique angle were incidenced so $\alpha$=10 ° was applied.

**R1C6-1)** 3. Parabolic bay shape equation [L164–180] >>> Renumbered as new Sect. 2.1?

**AC6-1)** That section was relocated to new Sect. 2.1.

**R1C6-2)** [OL 165] "… is provided by…" → "…defines the location of a point P $(R, \theta)$ on an embayed beach in static equilibrium by…"

**AC6-2)** We modified it as follow.

[ML 106-107] The parabolic bay shape equation (PBSE; Hsu and Evans, 1989) defines the location of a point P $(\theta, R)$ on an embayed beach in static equilibrium by

**R1C6-3)** [OL 166–180] All the "sin" and "cos" should not be italicized.

[OL 169] "…between the wave crest line (wave crest base line) and the line that passes through the …parallel to the shore base line" → "… between the wave crest base line (at the focus) and the tangential line at …on the shore base line;"

[OL 173] "…= 0 is satisfied to endure…" → "…= 1.0 (unity) to ensure…"

**AC6-3)** We modified it to the term recommended (ML 113, 117).

[L180] Eq. (9). Please show key interim steps that lead to this equation.

**AC6-4)** We modified the equation as follows: $\alpha_b(\theta) = \tan^{-1}\left(\frac{dy}{dx}\right) = \tan^{-1}\left(\frac{\sin\theta - \theta\cos\theta}{\cos\theta + \theta\sin\theta}\right)$

**R1C7-1)** 4. Analysis of shoreline change caused by oblique waves [L184–242] >>> Please re-number the section and change the heading as new Sect. 2.1 and Sect. 2.2

**AC7-1)** That section was relocated to new Sect. 2.1 and Sect. 2.2.

**R1C7-2)** [OL 199] Pease show interim steps leading to Eq. (10).

**AC7-2)** Eq. (10) was derived by comparing the longshore sediment transport rate and the amount of shoreline change on a surface area. We added the interim step of equation as follows: $\frac{Q_l t}{(h_c + h_B)} = \frac{x_c^2 \tan\alpha_b}{2}$

**R1C7-3)** [OL 199–237] All the "sin", "cos" and "cot" should not be italicized.

[L200–201] Unit "m3" and "m2" → superscript "m3" and "m2"

[L204] "Additionally, the time to reach static equilibrium, t1/2, when xc= L/2, is provided by" → "When static equilibrium is reached for xc= L/2, the time".

**AC7-3)** We modified it to the expression recommended (ML 168)

**R1C7-4)** [OL 206] "… wave power decreases…"??

**AC7-4)** We modified it to the term as follow: LST (ML 172)

**R1C7-5)** [OL 210] Please show key interim steps leading to Eq. (13).

**AC7-5)** We added the interim step of equation as follows: $\frac{y_{\pi/2}}{y_g} \cong \frac{y_g - R(\pi/2)}{y_g} = 1 - \frac{2}{\pi}\frac{\beta}{\sin\beta}$

**R1C7-6)** [OL 223–224] Please show key interim steps leading to Eqs. (14a) and (14b).

**AC7-5)** We added the interim step of equation as follows.

$$x_E = R(\theta)\sin(\varphi) = R(\theta)\sin\left(\frac{\pi}{2} + \alpha_b - \theta\right) \ / \ y_E = y_g - R(\theta)\cos(\varphi) = y_g - R(\theta)\cos\left(\frac{\pi}{2} + \alpha_b - \theta\right)$$

**R1C7-7)** [OL 241–242] Please consider to replot Fig. 14 using "$\alpha$" values of 0o, 2.5o, 5o, 7.5o, 10o, 15o   and 20o, for better viewing and manual application.

**AC7-7)** We modified the figure as recommended.

**R1C8-1)** 5. Results of comparison with numerical model [OL 243–272] >>> Re-number section as new Sect. 4 and also change section heading as: Results and comparison

5.1 Shoreline change model [OL 244–256] >>> First part in new Sect. 4.1: Analytical results with numerical results

**AC8-1)** That section was relocated to new Sect. 4 and Sect. 4.1.

**R1C8-2)** [OL 245–247] "…for the shoreline change model…sediment transport. It determines the shoreline change due to the difference in the LST along the coast within the active zone between the…" → "… for shoreline change…within a control volume due to the difference in LST across the active beach zone from berm…"

**AC8-2)** We modified it as follow.

[ML 355-357] The governing equation for shoreline change model (Pelnard–Considère, 1956) is a mass conservation equation of sediment transport. It determines the shoreline change due to the difference in the LST along the coast within the active zone between the berm and the depth of closure,

**R1C8-3)** [OL 248 Eq. 20; OL 250; OL 253 Eq. 21; OL255] Replace all four "$Q_y$" by "$Q_l$", to be consistent with the LST in the existing Sect. 2.2 (or new Sect. 3.2)?

**AC8-3)** We modified it to the expression recommended.

**R1C8-4)** [OL255–256] "…, we calculated or assigned the quantity of the LST at each grid. For example, we used $Q_y = 0$ for the eroding shoreline along the boundary of the groyne." → "…, the quantity of LST at each grid is calculated or assigned. For example, $Q'' = 0$ is assigned along the updrift groin where shoreline is receding."

**AC8-4)** We modified it as follow.

[ML 369-371] In the numerical calculations, the quantity of the LST at each grid is calculated or assigned. For example, $Q_l = 0$ is used for the eroding shoreline along the boundary of the groin.

**R1C9-1)** 5.2 Comparison between theoretical and numerical results [OL257–272] >>> Second part in new Sect. 4.1: Analytical results with numerical results

**AC9-1)** That section was relocated to new Sect. 4.1.

**R1C9-2)** [OL 269] "…gives $x_c$= -32.5 m, …"  →  "… gives $y_E$= - 32.5 m, …"?

**AC9-2)** We modified it as $y_E$= - 32.5 m (ML 387)

**R1C9-3)** [OL 270–272, Fig. 16] Replace the "Offshore (m)" on the ordinate by "Cross-shore (m)"

**AC9-3)** We modified the figure as recommended (Fig. 18).

**R1C10-1)** 6. Results of comparison with Jeongdongjin monitoring data [OL 273–317] >>> Renumber Sect. and heading as in new Sect. 4.2: Analytical results versus shoreline monitoring results

**AC10-1)** That section was relocated to new Sect. 4.2.

**R1C10-2)** [OL 274–276] "As shown in Fig. 17, where … to September 27th, 2013 to November 21st, 2016." >>> to be relocated after the new Fig. 10 at the end of new Sect. 3.1, and the two original sentences are revised as "Nearshore wave data were collected by an AWAC wave meter at a depth of 32.4 m to the south of Jeongdongjin Beach. From the data recorded over three years (27 September 2013 to 21 November 2016), the distribution of the annual mean wave direction is plotted (Fig. 11). As shown in Fig. 11, the prevailing wave direction was within -15° and +10° from the normal to the shoreline."

[OL 282–283] Original Fig. 17 to be re-numbered as new Fig. 9 and relocated under new Sect. 3.1.

**AC10-2)** We relocated the original Fig. 17 to Sect. 3 and Fig. 18 to Sect 3.1. And it has been modified as follows:

[ML 271-272] Nearshore wave data were also collected by an AWAC wave meter (Fig. 10) at a depth of 32.4 m to the south of Jeongdongjin Beach. From the data recorded over three years (27 September 2013 to 21 November 2016), distribution of the annual mean wave direction is plotted (Fig. 12). The results reveal the prevailing wave direction was mostly within -15o to +10o from the normal to the shoreline.

In addition, the relocated figure is attached below.

[Figure]

Fig. 10. Location of Jeongdongjin Beach in Gangwon-do on the east coast of South Korea. (Image from Google Earth)

**R1C10-3)** [OL 276–281] "As the distribution of the annual …obtaining fairly similar results." >> Suggestion: Please delete this part, as it may be irrelevant.

[OL 283–284] Original Fig. 18: The shoreline with gama-groin and legend may be deleted and just retain the little plot (called new Fig. 11) that shows the 'Distribution of the annual mean wave direction'.

[OL 285–286] Figure caption for the original Fig. 18 may be revised as:

"Fig. 11: Distribution of annual mean wave direction obtained from AWAC meter during 2013 – 2016."

**AC10-3)** We deleted the figure of gamma-groin as recommended. In addition, the relocated and modified figure is attached below.

[Figure]

Fig. 12. Distribution of the mean wave direction collected by an AWAC meter near Jeongdongjin Beach.

**R1C10-4)** [OL 287–290] "Shoreline monitoring in Korea…Project. The survey as also conducted to promote…based on scientific data accumulation and analysis; at Jeongdongjin, a video monitoring program that used four cameras…, covering 3,280 m (97.3%) of the total shoreline within a total of 3,370 m (Fig. 19)" → "Shoreline monitoring in Korea…Project, aiming to promote…, based on data collection and analysis. At Jeongdongjin, a video monitoring program employing four cameras…, which covers 3,280 m (97.3%) of a total of shoreline about 3,370 m (Fig. 18)."

Question: Are the values of "3,280 m" and "3,370 m" correct??

Please double check the correctness of the length of shoreline cited in the original MS [OL290], because the length of Jeongdongjin Beach is only about 800 m (see [OL330] in the original MS).

**AC10-4)** The total length of the Jeongdongjin's littoral cell is 3,370 m. Among them, 3,270 m of shores are covered by camera monitoring. The length of Jeongdongjin Beach is about 850 m. However, including the natural rocks, the lottoral cell on the north side was also included for monitoring, so it was mentioned to total 3,370 m. The content that was intended to be delivered was delivered incorrectly. So we modified it as follow.

[ML 397-398] At Jeongdongjin, the video monitoring program has commenced since February 2014, by installing four cameras to cover 3,280 m, which is the part (97.3 %) of the total littoral cell of 3,370 m including the northern part of the natural rocks (Fig. 19).

**R1C10-5)** [OL 293] "the continuously changing shoreline…" → "the spatial and temporal change in shoreline…"

**AC10-5)** We modified it as follow.

[ML 405] In this study, the variation of shoreline caused by the seasonal waves…

**R1C10-6)** [OL 295–303] Delete four "we"s in "we divided…", "we determined…", "we applied…" and 'we analyzed…"

[OL 294–296] "images, we divided the accumulative sum…every pixel by the number of captured images, from

which we determined the coordinates…and changing shoreline." → "images, the cumulative sum…of every pixel is divided by the number of the images captured, to determine the coordinates… and changing shoreline."

[OL 297] "image, we applied the geometric transformation equation of Lippmann and Holman (1989), which transforms…" → "image, the geometric transformation equation given by Lippmann and Holman (1989) is applied to transform…"

**AC10-6)** We modified it as follow.

[ML 406-408] To extract the average pixel value from the images, the cumulative sum of the attribute values of every pixel is divided by the total number of the images, from which the coordinates of the ground control points and the shoreline changes are assessed. The plane coordinate for the shoreline image is then rectified applying the geometric transformation equation (Lippmann and Holman, 1989), which transforms the image coordinates to ground coordinates as follows,

**R1C10-7)** [OL 299] The two "tan"s in Eq. (22a) should not be italicized.

**AC10-7)** We modified it as recommended.

**R1C10-8)** [OL 303–304] "By using this method, we analyzed the images of critical points taken twice a day from December 6 – 30, 2015, at Jeongdongjin Beach, as shown in Fig. 20, and compared them with the theoretical solution." → "Images of the critical points that were taken twice daily on Jeongdongjin Beach during December 6 – 30, 2015 are analyzed and compared with the analytical solution, as marked in Fig. 20."

**AC10-8)** We modified it as follow.

[ML 417-419] The video images were taken twice a day during December 6 – 30 in 2015, as shown in Fig. 20, and compared with analytical solution.

**R1C10-9)** [OL 307–308] "…, our results of the video…with those of the theoretical solution for the…) that used the PBSE approximation,…" → "…, the results of video…with that of the analytical solution for the…) predicted by the analytical approach, …"

**AC10-9)** We modified it as recommended (ML 421-422).

**R1C10-10)** [OL 309–311] "…video monitoring data… in November and December 2015, while we obtained the theoretical results from the analysis of the NOAA wave data within the same period of time." → "…video monitoring data … in December 2015, while the analytical results are calculated applying the NOAA wave data within the same period."

**AC10-10)** We modified it as follow.

[ML 421-422] In this figure, video monitoring data indicate the location of the critical points caused by seasonal oblique high waves in November and December in 2015, whilst the analytical results are obtained from the analysis of the NOAA wave data within the same period of the time.

**R1C11-1)** 6. Results of comparison with Jeongdongjin monitoring data [OL 318–333]

[OL 318–333] Please relocate this part to new Sect. 5: Discussions (1)

**AC11-1)** That section was relocated to new Sect. 5: Discussions (1).

**R1C11-2)** [OL 318] "Because our results exclude shoreline retreat because of cross-shore sediment transport, the theoretical solution…" → "Due to cross-shore sediment transport is excluded in the present study, the analytical solution…"

**AC11-2)** We modified it as follow.

[ML 443-444] Because the results presented in this study exclude the shoreline retreat due to cross-shore sediment movement, the analytical results for the maximum indentation from Eqs. (10a) and (10b) with LST might be underestimated.

**R1C11-3)** [OL 319–320] "…, both theoretical equations neglect…." → "…, the mathematical equations (i.e., Eqs. (22a) –(22b)) neglect…"

[OL 323] Eq. (23): "tan" should not be italicized.

**AC11-3)** We modified it as recommended (ML 445, 449).

**R1C11-4)** [OL 324–325] "Here, the subscript 1 denotes the limiting value. Table 2 compares the variables $x_c$ and $x_c^l$, obtained from …, respectively. If $x_c$ obtained for each $\alpha_b$ is greater than $x_c^l$ obtained a given $y_g$,"

→ "where subscript l denotes the limiting value. Variables $x_c$ and $x_c^l$, obtained from …, respectively, are compared in Table 2. If $x_c$ for each $\alpha_b$ is greater than $x_c^l$ for a given $y_g$,"

**AC11-4)** We modified it as follow.

[ML 451-452] Here, variables $x_c$ and $x_c^l$, obtained from Eqs. (10a, 10b) and (23), respectively, are compared in Table 2. If $x_c$ obtained for each $\alpha_b$ is greater than $x_c^l$ for a given $y_g$,

**R1C11-5)** [OL 327] "…the theoretical solution that the use of $\alpha_b = 10°$ for…" → "…the analytical solution that uses $\alpha_b = 10°$ for…"

**AC11-5)** We modified it as recommended (ML 454).

**R1C12)** 7. Discussions [OL 334–350] >>> Renumber this section as new "5. Discussions" [OL 334–350] Please relocate this part to new Sect. 5: Discussions (2)

• Please expand the Discussion section by including description on the limitations of the analytical approach presented in this study, as compared with other known theoretical methods and/or numerical models for predicting shoreline changes!

**AC12)** That section was relocated to new Sect. 5: Discussions (2).

Limitations: [443-444] Because the results presented in this study exclude the shoreline retreat due to cross-shore sediment movement, the analytical results for the maximum indentation from Eqs. (10a) and (10b) with LST might be underestimated as compared with methods and/or numerical models for predicting shoreline changes.

**R1C13)** 8. Conclusions [OL 351–376] >>> Renumber as new "6. Concluding remarks" [L351–376] Please revise.

**AC13)** That section was relocated and revised to new Sect. 6. Concluding remarks.

**R1C14)** References [OL 387–444]

Please double check the references, and remove redundant list. >>> Please add the following references:

Bakker, W.T., 1968. The influence of longshore variation of the wave height on the littoral current. Report WWK71-19, Ministry of Public Works, The Netherlands.

Balaji, R., Kumar, S.S., Misra, A., 2017. Understanding the effects of seawall construction using a combination of analytical modelling and remote sensing techniques: Case study of Fansa, Gujarat, India. J. Ocean and Climate Systems 8 (3), 153–160.

Ozasa, H., Brampton, A.H., 1980. Mathematical modeling of beaches backed by seawalls. Coastal Eng. 4 (1), 47–64.

**AC14)** We double check the references and removed redundant list. And we added the references.

---

## Author Comment (AC2)

**File AC – Response to both Referees' RC1 & RC2 on Esurf-2021-71**

**Preface:** **R#**: Referee number (1 or 2); **A**: Authors' response.
Line number in **Original MS**: [OL#]
Line number in **MARKED copy** and **FINAL MS**: [L#]

**Procedure of Revision**

(1) On 15/09/2021, authors submitted the original MS (Esurf-2021-71).

(2) On 30-31/10/2021, Referee 1 (R1) uploaded a comment file 'Esurf-2021-71-**RC1**'.

(3) On 24/11/2021, authors submitted response file to **RC1** called 'Esurf-2021-71-**AC1-supplement**'
   and produce a revised MS called '**Interim MS**', in which revision is marked in RED colour.

(4) On 24/11/2021 Referee 2 (R2) uploaded a comment file 'Esurf-2021-71-**RC2**'.

(5) Authors then undertake further revision and edit based on the "**Interim MS**". During this stage,
   several tasks are involved: (i) Restructuring Section and Sub-section headings (mentioned in A2 [3] below),
   (ii) Renumbering many equation numbers and figure numbers, (iii) Producing a new figure (i.e., Fig. 1), as well
   as deleting four original figures (Figs. 5, 10, 17 and 19) and Table 1, and (iv) Redrawing or modifying eight
   existing figures, for example, original Figs. 4, 7, 9, 14, 15, 18, 20 and 22 (now as new Figs. 11, 13, 4, 8, 15, 10,
   17 and 19).
   Consequently, some line numbers in the Interim MS have to be changed, and are not listed.

(6) Produce a combined response to R1 and R2 in **AC Response** 20211201 (**attached**)

(7) Produce a '**MARKED copy**' 20211201 (**attached**) in which the revised parts based on **CR2** are shown in
   YELLOW shading and that in the 'Interim MS' with RED colour.

(8) A clean '**FINAL MS**'20211201 (**attached**) is also produced by removing all colours in the **MARKED copy** (except
   that in Figures).

**General Comments from Referee 1 and Authors' Response**

**A. General Comments**

**R1-A1**). This referee has perused "ESurf-2021-71-v1".

**R1-A2**). This referee is familiar with the parabolic bay shape model that the authors adopted to derive analytical equations for estimating the location of the downdrift control point and the maximum indentation point within an eroded crenulate shoreline planform resulting from beach flanking at a natural groin. In the MS, the authors also validate their analytical solution with that from one-line numerical model and shoreline video monitoring data.

**R1-A3**). Like no others (e.g., Balaji et al., 2017; etc.) that apply merely the end results of R/Ro (i.e., relative radius distance from the updrift control point to a point on the static planform) from the parabolic model, these authors have extended  the applicability of the parabolic model by deriving new mathematical equations to analyze the spatial and temporal changes of the eroded bay shape. These equations are unique and useful for practical engineers to examining beach flanking shape — a phenomenon that has rarely been taught in the classrooms nor well documented in the literature, despite the erosion at downdrift end of seawalls and groins are common in field condition.

**R1-A4**). Application of their mathematical equation is straight forward, upon the input of long-term wave data (e.g., breaking wave height and angle, plus longshore sediment transport rate calculated

from the input wave condition). This analytical approach will also benefit the readers of Earth Surface Dynamics and other coastal engineering journals on studying beach erosion at the downdrift end of a land-based coastal structure (e.g., seawall or groin).

**R1-A5**). However, considering many research papers on beach erosion/change (e.g., using theoretical, analytical, experimental and numerical) have been published in learned journals and conference proceedings, the authors must state (in 1. Introduction) why they adopt an analytical approach based on the parabolic model for headland-bay beaches, rather by applying some of the existing methods available.

**R1-A6**). The quality of this paper has to be improved first, before the method described in the MS could become available to the readers, by overcoming the shortcomings in the current version of the MS. These include redundant words/phrases, readability and grammar in English, title of the paper, overall re-organization/revision of the contents, sections and sub-sections, as well as limitation of the proposed approach.

**A1 [A1-A6]** This paper attempted an analytical approach to beach flanking, which have occurred at several places on the east coast of South Korea. As noted in the general comments of Referee 1, this analytical approach will also benefit the readers of the Earth Surface Dynamics and other coastal engineering journals on studying beach erosion at the downdrift of structures that interrupt longshore sediment movement.

In order to improve the quality of this manuscript, these authors have undertaken a major revision, based on the comments of Referee 1. The task includes removing redundant words/phrases, improving readability and grammar in English, modifying the title of the paper, overall re-organising and revising the contents, section and sub-section headings.

**B. Specific comments**

**R1 [S1]** The current title only partially reflects the contents of the paper, and analysis, results and comparison are loosely spread in various sections and sub-sections. Integration should be considered.

**A1 [S1]** The paper has been revised with the title and contents as recommended.

**Original MS:**
   **Title:** "The sagging shape of shoreline formed on downdrift side of the structures due to seasonal oblique wave incidence"
   1. Introduction
   2. Analysis of seasonal incident waves
        2.1 Analysis of NOAA data
        2.2 Analysis of seasonal longshore sediment transport
   3. Parabolic bay shape equation
   4. Analysis of shoreline change caused by oblique waves
   5. Results of comparison with numerical model
        5.1 Shoreline change model
        5.2 Comparison between theoretical and numerical results
   6. Results of comparison with Jeongdongjin monitoring data
   7. Discussions: engineering countermeasures for mitigating seasonal erosion
   8. Conclusions

   to
   **Revised MS**:
   **Title**: "Analytical approach for beach flanking downdrift of natural groin induced by winter high waves: Case study of Jeongdongj in Korea"
        1. Introduction

**R1 [S2]** Abstract [OL 9–21]: Text written loosely. Please rewrite. Please also provide keywords.

**A1 [S2]** The Abstract is rewritten. It is now read as follows.

[L 9-19] Beach erosion at the unprotected downdrift end of a groin is common, when waves approaching obliquely to the structure. This phenomenon, known as beach flanking, has often occurred at downdrift of natural groins on the east coast of South Korea during high waves in winter months. The resulting planform assumes a distinctive crenulate shape with a maximum indentation point landward of the erosion. Analytical approach is employed to study the flanking at the downdrift end of a natural rock groin at Jeongdongjin Beach in Korea, using mathematical equations derived from the parabolic model for headland-bay beaches in static equilibrium, to predict the downdrift control point and maximum indentation of the eroded shoreline. These equations are solves using the prevailing wave height, wave angle at breaking and wave direction derived from analyzing NOAA's wave data over 40 years, and the longshore sediment transport rate calculated from the wave data. Location of the calculated maximum indentation is also verified using shoreline video monitoring data and compared with the result of one-line numerical model for shoreline change. Limitation of the proposed analytical approach is discussed, as well as the effect of sediment bypassing the groin.

[L 20] Keywords: Beach flanking, Analytical approach, Parabolic model, Maximum indentation point, Shoreline video monitoring.

**R1 [S3-1]** 1. Introduction [OL 23–78]

Please define "beach flanking" at the beginning of the Introduction and describe the effect of groin on sediment transport.

Please describe methods and/or models (numerical etc.) for analyzing beach erosion, and show references.

Please replace "sagging" by an appropriate word throughout.

**A1 [S3-1]** First of all, we adopt the term "beach flanking" rather than "sagging" in the entire MS. The term "beach flanking" was cited in Balaji et al. (2017), who applied the parabolic model to study the flanking at downdrift of seawalls in India.

[L 22–30] Although seawalls of vertical or sloping (revetments) have been used for many decades as a purported protection in an erosive situation, it is however unfortunate that they have often promoted further erosion, not only to the beach in front of them but also at downdrift, where a seaward-concave planform is produced (Kraus and McDougal, 1996). On the other hand, groins of moderate dimension running from the beach into the sea at right angles or inclined have also produced unwanted beach erosion, despite they were installed to intercept/accumulate sediment on the updrift side. This type of beach erosion at the downdrift

end of a shore-based coastal structures (e.g., seawalls and groins), which is known as beach flanking, is common, yet rarely being taught in the classrooms nor well documented in the literature. It results in a localized eroding beach in crenulate shape. In the case of groins, whilst the sediment being accreted, their downdrift beach that suffers erosion can only recover after the updrift shoreline has built up to the tip (head) of the structures, after sediment bypassing occurs.

**R1 [S3-2]** [OL 23] "… the eastern coast of …"  →  "… the east coast of …"?

[OL 31] "… a group of natural (pillar) rocks…"  →  "…a cluster of natural pillar rocks…"

[OL 44] "…for project planning."  →  "…for project planning (USACE, 2002)"

[OL 54] "…sagging shape…"  →  "…crenulate shape…"

**A1 [S3-2]** All the recommendations are implemented:
   [L 31] "… the east coast of …"
   [L 35-36] "…a cluster of natural pillar rocks…"
   [L 65] "…for project planning (USACE, 2002)"
   [L 63] "…crenulate shape…"

**R1 [S3-3]** [OL 36] "Many studies….". Please provide references.

**A1 [S3-3]** Several References are added to the list:
   [L 45-47] Beach erosion at downdrift of groins and their negative impact have been well studied theoretically or in prototype (Lehnfelt and Svendsen, 1958; Bakker, 1968, Bakker et al., 1970; Price and Tomlinson, 1970; Magoon and Edge, 1978; Headland et al., 2000; USACE, 2002).

**R1 [S3-4]** [OL 42] "the LST becomes static to maintain the shoreline planform…". Is "becomes static" correct?

**A1 [S3-4]** We revise it as follow:
   [L 62-63] When a SEP is reached, LST is not required to maintain the shoreline stability, because waves would break simultaneously along the bay periphery (Hsu et al., 2000).

**R1 [S3-5]** [OL 47] "To reproduce the shapes of shorelines in the laboratory,"  →  please revise.

**A1 [S3-5]** We revise it as follow.
   [L 43-44] Badei (1994) conducted laboratory experiments to reproduce the shoreline planform and topographic changes, while Wang and Kraus (2004) performed tests on erosion without longshore sediment transport (LST).

**R1 [S3-6]** [OL 56] "High resolution numerical models…". High resolution?

  **A1 [S3-6]** We revise it as recommended:.

  [L 68-69] … ,supported by numerical models (e.g., Xbeach and Sbeach) that include waves, currents and topographic changes.

**R1 [S3-7]** [OL 58–59] "These laboratory experiments … contributed to solving scientific equations about erosion by providing a similar sagging shape." What do you want to say? Please revise.

[OL60] "… still lack reliable factor analysis." What is 'factor analysis'? Is this a proper technical term?

**A1 [S3-7]** We delete it as recommended.

**R1 [S3-8]** [OL 61] "This study developed a theoretical approach, …". Past tense? 'develop'?

[OL 61–64] Please use verbs in present tense.

[OL 64–65] "…formed frequently…"?, "…a group of…"?

[OL 61–78] "theoretical approach", "theoretical analysis" and "theoretical solutions" etc. → Use "analytical approach", "analysis" and "analytical solutions"?

Please avoid repeating the use of some expressions, such as "Section 2…." at the beginning of a sentence or "…in Section X." at the end of a sentence.

[OL 77] "… through a theoretical formula that facilitates factor analysis."?? Please revise/clarify.

**A1 [S3-8]** We have almost revised the entire Introduction. Now it is read as:
> [L 70-81] The aims of this paper are threefold, (1) to derive mathematical equations for calculating the position of the maximum indentation in the eroded beach, (2) to demonstrate the applicability of an analytical approach derived from the parabolic model, and (3) to apply the mathematical equation to beach flanking at Jeongdongjin in Korea. These equations are solved using the prevailing wave conditions and the LST in winter months at Jeongdongjin which are obtained from analyzing NOAA's wave data. In this paper, a brief introduction is first given in section 1, while section 2 describes the analytical approach using the parabolic model and the derivation of mathematical equations for the downdrift control point and the maximum indentation on the eroded beach. Analysis of NOAA's wave data over 40 years is presented in section 3.1 which provides averaged wave heights and wave angles at breaking for solving the mathematical equations (sections 2.2 and 2.3) and the seasonal LST rate (section 3.2) for one-line shoreline change model outlined in section 4.1. The analytical results for the maximum indentation point are then compared with the results of the numerical model and that from shoreline video monitoring project at Jeongdongjin Beach (section 4.2). Finally, discussions on the limitation of the proposed analytical approach and the effect of sediment bypassing are given in section 5. Concluding remark is given in section 6.

**R1 [S4-1]** 2. Analysis of seasonal incidence waves [OL 79] → please change section heading.

**A1 [4-1]** The title of the section heading was specifically modified as follow.
> Revised section heading: "Wave and Shoreline Monitoring Data for Jeongdongjin Beach"

**R1 [S4-2]** 2.1 Analysis of NOAA data [OL 80–99] >>> Relocate this section to new Sect. 3.1?

**A1 [S4-2]** This section is relocated to new Sect. 3.1.

**R1 [S4-3]** [OL 86; OL 97; OL 99] "eastern coast" → "east coast"?

**A1 [S4-3]** We have revised it as recommended.

**R1 [S4-4]** [OL 88] "…cause by the energy of the oblique high waves."? Please revise.

**A1 [S4-4]** We have revised it as recommended.
> [L 212-213] The wave data are analyzed and the results used to calculate the change of the eroding shoreline curve at downdrift of the natural rocky groin caused by the oblique high waves in the winter.

**R1 [S4-5]** [OL 95–96] "…Average over 10 year-intervals..." → "…Averaged over every 10- year period…"?

[OL 97–98] "…local shoreline orientation in Gangwon-do was approximately N43°E." → "…local shoreline orientation at Jeongdongjin is in WN to SE direction, or N133°E."?

**A1 [S4-5]** We modified it to the term recommended.

    [L 218] … averaged over every 10-year intervals …

    [L 220] … and the local shoreline aligns in NW – SE direction (about N133°E).

**R1 [S5-1]** 2.2 Analysis of seasonal longshore sediment transport [OL100–160] >>> Relocate entire section to new Sect. 3.2?

**A1 [S5-1]** This section ias relocated to new Sect. 2.2.

**R1 [S5-2]** [OL 101–146] All "$Q_y$", "$P_y$", "$I_y$" → "$Q_l$", "$P_l$", "$I_l$"? because subscript "y" is not defined, while subscript "l" is the most used term in textbook.

All the "sin" and "cos" should not be italicized.

[OL118] "…the isobath of the seabed…" → "… seabed contours…"?

**A1 [5-2]** We modified it to the term recommended.

**R1 [S5-3]** [OL 121] How to derive Eq. (5), reference?

**A1 [S5-3]** Eq. (5) is derived using longshore sediment transport equation (see the book of Dean and Dalrymple (2001) : Coastal Processes with Engineering Applications, "Littoral Drift Computations based on Deep Water Data" parts) and is not cited separately.

**R1 [S5-4]** [OL133–138] Paragraph and the original Fig. 5 may be deleted?

**A1 [S5-4]** We delete them as recommended.

**R1 [S5-5]** [OL 151–152] "…high waves in winter… arrives from…N38°E-2.5° to N38°E +7.5°" >> but $\alpha_b$ = 10° is used in Figs. 16 and 20, Why?

**A1 [S5-5]** The case of $\alpha_b$=10° is only one of the example, not the most appropriate one. We have revised the paragraph [L 284-289], stating that "First, the results of the numerical model for a series of time duration from 6 hours to 4 weeks can be first demonstrated, using $\alpha_b$ = 10° (Fig. 13), are indicated in Fig. 16, …. In addition, the numerical model is also run for other wave angles at breaking ($\alpha_b$) at 2° intervals, and the results are collectively marked as "Analytical results" in Fig. 17."

**R1 [S6-1]** 3. Parabolic bay shape equation [L164–180] >>> Renumbered as new Sect. 2.1?

**A1 [S6-1]** This section is relocated to new Sect. 2.1.

**R1 [S6-2]** [OL 165] "… is provided by…" → "…defines the location of a point P ($R$, $\theta$) on an embayed beach in static equilibrium by…"

**A1 [S6-2]** We revise it as follow.

    [L 84-85] The parabolic bay shape equation (PBSE; Hsu and Evans, 1989) defines the location of a point P ($\theta$, $R$) on an embayed beach in static equilibrium by

**R1 [S6-3]** [OL 166–180] All the "sin" and "cos" should not be italicized.

[OL 169] "…between the wave crest line (wave crest base line) and the line that passes through the …parallel to the shore base line" → "… between the wave crest base line (at the focus) and the tangential line at …on the shore base line;"

[OL 173] "…= 0 is satisfied to endure…" → "…= 1.0 (unity) to ensure…"

**A1 [S6-3]** We revise it as recommended (L 86-87, 93).

**R1 [S6-4]** [L180] Eq. (9). Please show key interim steps that lead to this equation.

**A1 [6-4]** We insert the interim step to this equation: $\alpha_b(\theta) = \tan^{-1}\left(\frac{dy}{dx}\right) = \tan^{-1}\left(\frac{\sin\theta - \theta\cos\theta}{\cos\theta + \theta\sin\theta}\right)$

**R1 [S7-1]** 4. Analysis of shoreline change caused by oblique waves [L184–242] >>> Please re-number the section and change the heading as new Sect. 2.1 and Sect. 2.2

**A1 [S7-1]** This section is relocated to new Sect. 2.1 and Sect. 2.1.1.

**R1 [S7-2]** [OL 199] Pease show interim steps leading to Eq. (10).

**A1 [S7-2]** Now Eq. (4.2) was derived by comparing the longshore sediment transport rate and the amount of shoreline change on a surface area. We added the interim step of equation as follows: Eq. (4.1) $\frac{Q_l t}{(h_c + h_B)} = \frac{x_c^2 \tan\alpha_b}{2}$

**R1 [S7-3]** [OL 199–237] All the "sin", "cos" and "cot" should not be italicized.

[L200–201] Unit "m3" and "m2" → superscript "m3" and "m2"

[L204] "Additionally, the time to reach static equilibrium, t1/2, when xc= L/2, is provided by" → "When static equilibrium is reached for xc= L/2, the time".

**A1 [S7-3]** We modify the expression as recommended (L 127)

**R1 [S7-4]** [OL 206] "… wave power decreases…"??

**A1 [7-4]** We modified it to the term as follow: Eq. (6) implies that the time required to reach equilibrium increases as the LST decreases, but as beach length and wave obliquity increase. (L 130-131)

**R1 [S7-5]** [OL 210] Please show key interim steps leading to Eq. (13).

**A1 [S7-5]** We insert the interim step of this equation as follows: $\frac{y_{\pi/2}}{y_g} \cong \frac{y_g - R(\pi/2)}{y_g} = 1 - \frac{2}{\pi}\frac{\beta}{\sin\beta}$

**R1 [S7-6]** [OL 223–224] Please show key interim steps leading to Eqs. (14a) and (14b).

**A1 [S7-6]** We insert the interim step of this equation as follows.

$x_E = R(\theta)\sin(\varphi) = R(\theta)\sin\left(\frac{\pi}{2} + \alpha_b - \theta\right)$ / $y_E = y_g - R(\theta)\cos(\varphi) = y_g - R(\theta)\cos\left(\frac{\pi}{2} + \alpha_b - \theta\right)$

**R1 [S7-7]** [OL 241–242] Please consider to replot Fig. 14 using "$\alpha$" values of 0o, 2.5o, 5o, 7.5o, 10o, 15o and 20o, for better viewing and manual application.

**A1 [S7-7]** We modify the figure as recommended.

**R1 [S8-1]** 5. Results of comparison with numerical model [OL 243–272] >>> Re-number section as new Sect. 4 and also change section heading as: Results and comparison

5.1 Shoreline change model [OL 244–256] >>> First part in new Sect. 4.1: Analytical results with numerical results

**A1 [S8-1]** This section was relocated as new Sect. 4 and Sect. 4.1.

**R1 [S8-2]** [OL 245–247] "…for the shoreline change model…sediment transport. It determines the shoreline change due to the difference in the LST along the coast within the active zone between the…" → "… for shoreline change…within a control volume due to the difference in LST across the active beach zone from berm…"

**A1 [S8-2]** We revise the sentence as:

[ML 360-362] The governing equation for shoreline change model (Pelnard–Considère, 1956) is a mass conservation equation of sediment transport. It determines the shoreline change due to the difference in the LST along the coast within the active zone between the berm and the depth of closure,

**R1 [S8-3]** [OL 248 Eq. 20; OL 250; OL 253 Eq. 21; OL255] Replace all four "$Q_y$" by "$Q_l$", to be consistent with the LST in the existing Sect. 2.2 (or new Sect. 3.2)?

**A1 [S8-3]** We modified it to the expression recommended.

**R1 [S8-4]** [OL255–256] "…, we calculated or assigned the quantity of the LST at each grid. For example, we used $Q_y = 0$ for the eroding shoreline along the boundary of the groyne." → "…, the quantity of LST at each grid is calculated or assigned. For example, $Q'' = 0$ is assigned along the updrift groin where shoreline is receding."

**A1 [S8-4]** We modified it as follow.

[L 273-275] In the numerical calculations, the quantity of the LST at each grid is calculated or assigned. For example, $Q_l = 0$ is used for the eroding shoreline along the boundary of the groin.

**R1 [S9-1]** 5.2 Comparison between theoretical and numerical results [OL257–272] >>> Second part in new Sect. 4.1: Analytical results with numerical results

**A1 [S9-1]** This section is relocated as new Sect. 4.1.

**R1 [S9-2]** [OL 269] "…gives $x_c = -32.5$ m, …" → "… gives $y_E = -32.5$ m, …"?

**A1 [S9-2]** We correct it as $y_E = -32.5$ m (L 289)

**R1 [S9-3]** [OL 270–272, Fig. 16] Replace the "Offshore (m)" on the ordinate by "Cross-shore (m)"

**A1 [S9-3]** We modify the figure as recommended (Fig. 18).

**R1 [S10-1]** 6. Results of comparison with Jeongdongjin monitoring data [OL 273–317] >>> Renumber Sect. and heading as in new Sect. 4.2: Analytical results versus shoreline monitoring results

**A1 [S10-1]** This section is relocated to new Sect. 4.2.

**R1 [S10-2]** [OL 274–276] "As shown in Fig. 17, where … to September 27th, 2013 to November 21st, 2016." >>> to be relocated after the new Fig. 10 at the end of new Sect. 3.1, and the two original sentences are revised as "Nearshore wave data were collected by an AWAC wave meter at a depth of 32.4 m to the south of Jeongdongjin Beach. From the data recorded over three years (27 September 2013 to 21 November 2016), the distribution of the annual mean wave direction is plotted (Fig. 11). As shown in Fig. 11, the prevailing wave direction was within -15° and +10° from the normal to the shoreline."

[OL 282–283] Original Fig. 17 to be re-numbered as new Fig. 9 and relocated under new Sect. 3.1.

**A1 [S10-2]** We renumber the original Fig. 17 to Sect. 3 and Fig. 18 to Sect 3.1, as well as revise the sentences .

[L 223-227] Nearshore wave data were also collected by an AWAC wave meter (Fig. 10) at a depth of 32.4 m to the south of Jeongdongjin Beach. From the data recorded over three years (27 September 2013 to 21 November 2016), distribution of the annual mean wave direction is plotted (Fig. 12). The results reveal the

prevailing wave direction was mostly within -15 to +10 from the normal to the shoreline.

In addition, the renumbered figure is attached below.

[Figure]

**R1 [S10-3]** [OL 276–281] "As the distribution of the annual …obtaining fairly similar results." >> Suggestion: Please delete this part, as it may be irrelevant.

[OL 283–284] Original Fig. 18: The shoreline with gama-groin and legend may be deleted and just retain the little plot (called new Fig. 11) that shows the 'Distribution of the annual mean wave direction'.

[OL 285–286] Figure caption for the original Fig. 18 may be revised as:

"Fig. 11: Distribution of annual mean wave direction obtained from AWAC meter during 2013 – 2016."

**A1 [S10-3]** We deleted the figure of gamma-groin as recommended. In addition, the renumber and modify figure as attached below.

[Figure]

**R1 [S10-4]** [OL 287–290] "Shoreline monitoring in Korea…Project. The survey as also conducted to promote…based on scientific data accumulation and analysis; at Jeongdongjin, a video monitoring program that used four cameras…, covering 3,280 m (97.3%) of the total shoreline within a total of 3,370 m (Fig. 19)" → "Shoreline monitoring in Korea…Project, aiming to promote…, based on data collection and analysis. At Jeongdongjin, a video monitoring program employing four cameras…, which covers 3,280 m (97.3%) of a total of shoreline about 3,370 m (Fig. 18)."

Question: Are the values of "3,280 m" and "3,370 m" correct??

Please double check the correctness of the length of shoreline cited in the original MS [OL290], because the length of Jeongdongjin Beach is only about 800 m (see [OL330] in the original MS).

**A1 [S10-4]** The total length of the Jeongdongjin's littoral cell is 3,370 m. Among them, 3,270 m of shores are

covered by camera monitoring. The length of Jeongdongjin Beach is about 850 m. However, including the natural rocks, the littoral cell on the north side was also included for monitoring, so it was mentioned to total 3,370 m.

**R1 [S10-5]** [OL 293] "the continuously changing shoreline…" → "the spatial and temporal change in shoreline…"

**A1 [S10-5]** We modified it as follow.

[L 256] In this study, the variation of shoreline caused by the seasonal waves…

**R1 [S10-6]** [OL 295–303] Delete four "we"s in "we divided…", "we determined…", "we applied…" and 'we analyzed…"

[OL 294–296] "images, we divided the accumulative sum…every pixel by the number of captured images, from which we determined the coordinates…and changing shoreline." → "images, the cumulative sum…of every pixel is divided by the number of the images captured, to determine the coordinates… and changing shoreline."

[OL 297] "image, we applied the geometric transformation equation of Lippmann and Holman (1989), which transforms…" → "image, the geometric transformation equation given by Lippmann and Holman (1989) is applied to transform…"

**A1 [S10-6]** We modified it as follow.

[L 256-257] In this study, the variation of shoreline positions were derived from the time-averaged video images by the geometric transformation equation (Lippmann and Holman, 1989), which transforms the image coordinates to ground coordinates.

**R1 [S10-7]** [OL 299] The two "tan"s in Eq. (22a) should not be italicized.

**A1 [S10-7]** We modified it as recommended.

**R1 [S10-8]** [OL 303–304] "By using this method, we analyzed the images of critical points taken twice a day from December 6 – 30, 2015, at Jeongdongjin Beach, as shown in Fig. 20, and compared them with the theoretical solution." → "Images of the critical points that were taken twice daily on Jeongdongjin Beach during December 6 – 30, 2015 are analyzed and compared with the analytical solution, as marked in Fig. 20."

**A1 [S10-8]** We modified it as follow.

[L 295-299] The video images were taken twice a day during December 6 – 30 in 2015, as shown in Fig. 20, and compared with analytical solution.

**R1 [S10-9]** [OL 307–308] "…, our results of the video…with those of the theoretical solution for the…) that used the PBSE approximation,…" → "…, the results of video…with that of the analytical solution for the…) predicted by the analytical approach, …"

**A1 [S10-9]** We modified it as recommended.

**R1 [S10-10]** [OL 309–311] "…video monitoring data… in November and December 2015, while we obtained the theoretical results from the analysis of the NOAA wave data within the same period of time." → "…video monitoring data … in December 2015, while the analytical results are calculated applying the NOAA wave data within the same period."

**A1 [S10-10]** We modified it as follow.

[L 295-296] The video images are compared with the analytical solution. Figure 17 shows the results of the video monitoring data are in fair agreement with that of the analytical solution for the maximum indentation points using the PBSE approximation.

**R1 [S11-1]** 6. Results of comparison with Jeongdongjin monitoring data [OL 318–333]

[OL 318–333] Please relocate this part to new Sect. 5: Discussions (1)

**A1 [S11-1]** That section was relocated to new Sect. 5: Discussions (1).

**R1 [S11-2]** [OL 318] "Because our results exclude shoreline retreat because of cross-shore sediment transport, the theoretical solution…" → "Due to cross-shore sediment transport is excluded in the present study, the analytical solution…"

**A1 [S11-2]** We modified it as follow.

[L 313-314] Because the results presented in this study exclude the shoreline retreat due to cross-shore sediment movement, the analytical results for the maximum indentation from Eqs. (10a) and (10b) with LST might be underestimated.

**R1 [S11-3]** [OL 319–320] "…, both theoretical equations neglect…." → "…, the mathematical equations (i.e., Eqs. (22a) –(22b)) neglect…"

[OL 323] Eq. (23): "tan" should not be italicized.

**A1 [S11-3]** We modified it as recommended.

**R1 [S11-4]** [OL 324–325] "Here, the subscript 1 denotes the limiting value. Table 2 compares the variables $x_c$ and $x_c^l$, obtained from …, respectively. If $x_c$ obtained for each $\alpha_b$ is greater than $x_c^l$ obtained a given $y_g$,"

→ "where subscript l denotes the limiting value. Variables $x_c$ and $x_c^l$, obtained from …, respectively, are compared in Table 2. If $x_c$ for each $\alpha_b$ is greater than $x_c^l$ for a given $y_g$,"

**A1 [S11-4]** We modified it as follow.

[L 319-320] Here, variables $x_c$ and $x_c^l$, obtained from Eqs. (10a, 10b) and (22), respectively, are compared in Table 1. If $x_c$ obtained for each $\alpha_b$ is greater than $x_c^l$ for a given $y_g$, then $x_c$ should be replaced by $x_c^l$ ,

**R1 [S11-5]** [OL 327] "…the theoretical solution that the use of $\alpha_b$ = 10° for…" → "…the analytical solution that uses $\alpha_b$ = 10° for…"

**A1 [S11-5]** We modified it as recommended (L 322).

**R1 [S12]** 7. Discussions [OL 334–350] >>> Renumber this section as new "5. Discussions" [OL 334–350] Please relocate this part to new Sect. 5: Discussions (2)

• Please expand the Discussion section by including description on the limitations of the analytical approach presented in this study, as compared with other known theoretical methods and/or numerical models for predicting shoreline changes!

**A1 [S12]** This section is renumbered as new Sect. 5: Discussions (2).

Limitations: [L 313-315] Because the results presented in this study exclude the shoreline retreat due to cross-shore sediment movement, the analytical results for the maximum indentation from Eqs. (10a) and (10b) with LST rate might be underestimated. In addition, both analytical equations neglect sediment bypassing from updrift.

**R1 [S13]** 8. Conclusions [OL 351–376] >>> Renumber as new "6. Concluding remarks" [L351–376] Please revise.

**A1 [S13]** This section is renumbered and revised as new Sect. 6. Concluding Remarks.

**R1 [S14]** References [OL 387–444]

Please double check the references, and remove redundant list. >>> Please add the following references:

Bakker, W.T., 1968. The influence of longshore variation of the wave height on the littoral current. Report WWK71-19, Ministry of Public Works, The Netherlands.

Balaji, R., Kumar, S.S., Misra, A., 2017. Understanding the effects of seawall construction using a combination of analytical modelling and remote sensing techniques: Case study of Fansa, Gujarat, India. J. Ocean and Climate Systems 8 (3), 153–160.

Ozasa, H., Brampton, A.H., 1980. Mathematical modeling of beaches backed by seawalls. Coastal Eng. 4 (1), 47–64.

**A1 [S14]** We double check the References and removed redundant list, as well as adding new References.

**General Comments from Referee 2 and Authors' Response**

Authors present a method based on the parabolic static equilibrium beach to estimate the main dimensions of the expected shoreline indentation downdrift groins. The method is then compared against numerical model results and applied to a beach in Korea. The manuscript addresses a topic covered by Earth Surface Dynamics and, in this sense, the manuscript can be of interest for many ESurfD readers.

**General comments**

**R2 [1]** The manuscript needs a thorough revision of the **English language**. Grammatical errors (and unusual sentence constructions) are very frequent throughout the manuscript and will not be indicated here except in selected cases.

**A2 [1]** Thank you. We have improved the readability of the manuscript.

**R2 [2]** The **title** is long and confusing. Please make it simpler. "Sagging shape" is not usually employed in the context of the topic, it is usually referred as shoreline indentation or erosion.

**A2 [2]** Thank you. We have revised the title to reflect the nature of the work reported in this manuscript, from "The sagging shape of shoreline formed on the downdrift side of the structures due to seasonal oblique wave incidence" (19 words) to "Analytical approach for beach flanking downdrift of natural groin: case study of Jeongdongjin Beach, Korea". (15 words)

**R2 [3]** Throughout the manuscript you mix data, methods and results in the same sections. This is a bit confusing for readers. Please re-organise the manuscript to include the following sections and restrict the content to the parts corresponding to the heading:
(i) **Study area and data** where you describe the study site characteristics and present all data to be used;
(ii) **Methodology** where you present and describe all methodology used in the study;
(iii) **Results** where all results are presented;
(iv) **Discussion**, where you discuss the obtained results and the applicability of the presented approach/model; and
(v) **Conclusions**.

**A2 [3]** Thank you. We have restructured the manuscript with appropriate headings and made clear separation of the methods, data and results, from
**Original MS:**
  1. Introduction
  2. Analysis of seasonal incident waves
    2.1 Analysis of NOAA data
    2.2 Analysis of seasonal longshore sediment transport
  3. Parabolic bay shape equation
  4. Analysis of shoreline change caused by oblique waves
  5. Results of comparison with numerical model
    5.1   Shoreline change model
    5.2   Comparison between theoretical and numerical results
  6. Results of comparison with Jeongdongjin monitoring data
  7. Discussions: engineering countermeasures for mitigating seasonal erosion
  8. Conclusions

  To become in
**Revised MS** now is reorganized as:
  1.   Introduction
  2.   Methodology
    2.1   Parabolic bay shape model

**R2 [4] Description of methods.**

At its present version, some of the text included in the description of the methods is "excessive". For instance, most of text describing the longshore sediment transport formula can be just simplified in a single line stating which is the formula to be used plus the formula itself. Most of this text can be found in any textbook. Please revise and simplify when possible. In some cases, you provide more information than strictly needed. For instance, in section 6 the text from line 294 to 303 can be summarise in something like "shoreline positions were derived from time-averaged video images by using the method proposed by X". Equation 22 is not needed.

**A2 [4]** Thanks for the suggestion.

(1)  On longshore sediment transport (LST) equation:

We opt to retain most part of Sect. 2.2, but delete [OL 132-238] and Fig. 5.

Although the LST equation is well known, we believe its inclusion can readily provide benefit to some readers, especially the additional form using deepwater wave conditions to express $Q_l$ derived from NOAA's wave data, and the plotting of $Q_l/C_{ol}$ versus Month (Old Fig. 6, now new Fig. 12).

(2)  One process of shoreline video monitoring images: As suggested, we have deleted the description on [OL 294-303] and Eq. 22.

**R2 [5] Introduction**

In this section you mix a series of concepts related to shoreline development. However, it is not clear why you do it. It would be better for readers if you clearly motivate your study.

**A2 [5]** The original 1. Introduction [OL 22-78] is rewritten, based on the comments of Referee 2. The revised Introduction [L 23-83] includes additional references and a new figure (i.e., Fig. 1), and the aim of this paper is also clearly stated.

**R2 [L 61-65]** Please, reformulate the paragraph and include a sentence where you explicitly state the objective of the paper. Something like "The main aim of this work is ……."

**A2 [L 61-65]** The main aim of this study (paper) is now clearly stated [L 72-83].

**R2 [L 147 – 152 & Fig 7]** How did you calculate $H_b$? How did you obtain $\alpha_b$?

**A2 [L 147 – 152 & Fig 7]** We now explain the method for estimating $H_b$ and $\alpha_b$ from NOAA's wave data offshore, as "Assuming the seabed contours are straight and parallel, then the wave height and wave angle at the breaker are estimated from satisfying linear wave shoaling and refraction relationship ($H/H_0 = K_s K_R$) and spilling wave breaking criteria ($h_b = H_b/\gamma$) simultaneously (Reeve et al., 2012) from the waves offshore collected in NOAA's wave data." [L 240–242]

**R2 [L 204-205]**

Define L (groin spacing / bay length)

**A2 [L 204-205]** The length of shoreline at Jeongdongjin is about 850 m, spanning between the cluster of pillar rocks and the artificial land-based structure (see Fig. 1).

**R2 [line 208]** *parallel shoreline approximation* of Hsu & Evans?

**A2 [L 208]** Sorry. There is no such thing as "parallel shoreline approximation". This is rewritten as "the tangent passing through the downdrift control point ($Q$) is parallel to the wave crest baseline" [L 91-92], as well as shown in the definition sketch (original Fig. 9 or new Fig. 4).

**R2 [L 252-253]** If you are going to propose an alternative expression for LST in the diffraction zone you have to consider not only changes in the wave angle, as you do in equation (21), but you should also include a term to account for currents induced by the wave height gradient occurring in the diffraction zone (e.g. Osaza & Brampton'1980 approach).

**A2 [L 252-253]** Lim et al (2021) have proposed an alternative form for the diffracted zone, as explained on the revised manuscript [L 275-278], stating "An alternative expression for the LST rate, similar to Eq. (17), is given by $Q_l = CH_b^{5/2} \sin 2\alpha_m$, (21) where $\alpha_m$ is the wave angle within the diffracted zone (Lim et al., 2021). However, nearshore currents within the diffraction zone is assumed to be in non-existent, when SEP is reached for the eroded shoreline planform."

**R2 [L 263]** why is $\alpha_b$=10° the most appropriate angle?

**A2 [L 263]** The case of $\alpha_b$=10° is only one of the example, not the most appropriate one. We have revised the paragraph [L 289-296], stating that "First, the results of the numerical model for a series of time duration from 6 hours to 4 weeks can be first demonstrated, using $\alpha_b$ = 10° (Fig. 13), are indicated in Fig. 16, …. In addition, the numerical model is also run for other wave angles at breaking ($\alpha_b$) at 2° intervals, and the results are collectively marked as "Analytical results" in Fig. 17."

**R2 [L 266-267]** Fig 16 does not indicate any agreement with field observations. Fig 16 just shows shoreline evolution as predicted by the model.

**A2 [L 266-267]** The original Fig. 16 is now the same number as new Fig. 16. We now delete this sentence, because the statement is misplaced under the sub-heading of '4.1 Comparison between analytical and numerical results'. Instead, we compare the temporal variation of the maximum indentation ($x_E, y_E$) as $\alpha_b$ increases from 10° to 20°. The explanation is given as "Figure 15 compares the temporal variation of the results for analytical and numerical results of $x_E$ and $y_E$ at the maximum indentation for $\tau$ ($= x_c^2/q_b$; unit in meter and hour) for three different values of $\alpha_b$ (10°, 15° and 20°) at Jeongdongjin Beach with a natural rocky groin about 80 m long. In the numerical results, the value of $y_E$ (negative) increases (erosion) with time as $\alpha_b$ increases, whilst that of $x_E$ decreases (closer to the boundary from the groin). Similar trend can be found in the analytical results. Moreover, the discrepancies between the numerical and analytical results increases as $\alpha_b$ increases." [L281-285]

**R2 [L 269]** I think that the $x_c$ = -32.5 is wrong, probably you refer to another thing and not to $x_c$.

**A2 [L 269]** Sorry, we are wrong. It is now corrected as $y_E$ = − 32.5 m [L 294].

**R2 [line 274]** Fig 17 is only a map where the location of AWAC is shown. It does not provide any crucial information (it should be enough to say that wave data were recorded by an AWAC system southward of the study area).

**A2 [L 274]** The original Figs. 17 and 19 are is deleted. Together, they are combined into the new Fig. 1, with the AWAC (in the original Fig. 17) and cameras (in the original Fig. 19) marked in the same picture.

**R2 [L 277-281]** Can you explain better what did you do? What do you refer with average shoreline? Which diffraction model?

**A2 [L 277-281]** The description of the equilibrium shoreline planform [original L 277-281] under the original section heading '6. Results of comparison with Jeongdongjin monitoring data' is deleted, as well as the shoreline part of in the original Fig. 18. Only the distribution of the annual mean wave direction is retained (now as new Fig. 11). However, the first part under this section that refers to AWAC and mean wave direction is now placed under wave data (Sect. 3.1) [L 225-231].

**R2 [L 307-313]** Here you are comparing nearly instantaneous critical points detected from hourly/daily video observations with predictions done for average seasonal conditions. Is this consistent to say that data agree?

**A2 [L 307-313]** This is explained in the revised MS as "The results of analytical method predict the eroding shoreline planform and the maximum indentation point, using the parabolic model with monthly averaged wave conditions from NOAA wave data, whereas that of the video monitoring program reveal the hourly/daily record from the shoreline images corresponding to the instantaneously changing wave conditions. To compare the results coming from these two different sources, the maximum indentation points occurred on the day of video recording is selected. Despite the difference in time scale, Fig. 17 indicate that the results of the analytical prediction are in fair agreement with the video monitoring data." [L 300–305]

**R2 [L 324]** subscript -> superscript

**A2 [L 324]** Yes, it should be "superscript" for the "$l$" in $x_c^l$. It is now corrected. [L 330].

**R2 [Discussion]**
Here, you don't formally discuss your work (neither the shortcomings of the methodology or the results obtained). You should include here the advantages and disadvantages of the method. Why am I going to use your approach?

**A2 [Discussion]** "5. Discussion" is rewritten/reorganized.

In addition to the existing part of mitigating beach flanking using beach nourishment and installation of an interim groin [OL 335–350], the new Section 5. Discussions [L 392–442] now includes (1) the advantage of using the method presented in the paper and modification required if applied to other places, (2) discussion on the effect of sediment bypassing, which was described in the original MS [OL 314–358].

**R2 [Conclusions]**
This section is too large (it is larger than the Discussion section). It is more a summary than conclusions.

**A2 [Conclusions]** Section 6. Concluding Remark [L 361–376] is now shortened

**R2 [Figure 2]**. please mark the location of the study site in the photo of Korea.

**A2 [Figure 2]** The study site is now marked in the new Fig. 1.

**R2 [Figure 10 and table 1**] are providing the same info. Also, they have been obtained by applying equation 9 to selected θ values. You could remove them (figure and table) without affecting the understanding of the text. In any case, please remove at least one of them (table or figure).

**A2 [Figure 10 and table 1**] Thanks for suggestion. The original Fig. 10 and Table 1 that bear identical results are deleted. Instead, a new sentence of "The relationship between $\theta$ and $\alpha_b$ in Eq. (3) for a SEP can be readily calculated and expressed explicitly using figure or table. For example, the three key values of $\alpha_b$ (= 10º, 20º and 30º) correspond to $\theta$ of 52.5º, 71.2º and 86.5º, respectively." are noted. [L 104-105]

**R2 [Figure 19]** is not strictly needed.

**A2 [Figure 19]** Thanks for suggestion. The image of cameras is now indicated in new Fig. 1.

---

## Referee Report (RR1)

**Final report of Referee 1 on revised ESurf-2021-71**

**A. General Comments**

**A1**). This referee has perused the Marked Copy, called "esurf-2021-71-ATC1", for the revised MS of ESurf-2021-71, together with authors' response to both referees (RC1 and RC2) as documented in "esurf-2021-71-author_response-version1".

**A2**). In the authors' response file, it can be found that many similar comments were given by both referees, especially on the issues related to (1) English grammar and readability of the MS, (2) title of the paper, (3) overall organizational structures (sections and sub-sections) of the presentation, (4) confused mix of analyses, results and discussions within the same section, and (5) redundant figures, etc.

**A3**). From comparing the original MS with the Marked Copy that yields the Clean Copy "esurf-2021-71-manuscript-version3" — the final MS, this referee believes that the authors have satisfactorily revised the MS, based on the comments raised by both referees. The results show: (1) the revised title is adequate to reflect the context of the MS, (2) readability and grammar have improved, (3) overall structures of the presentation (especially sections and sub-sections) well revised and organized, (4) Abstract is rewritten, with revised Introduction and additional references, (5) clear separation is made between analysis and results, and (6) redundant figures are removed and new figures appended.

**A4**). In the MS, the authors have adopted the term "beach flanking" to replace the original term of "sagging shoreline". This referee agrees with the authors' statement that "… beach flanking, is common, yet rarely being taught in the classrooms nor well documented in the literature."

**A5**). On the application of the parabolic model to crenulate beaches, like no others that apply only the end results of $R/R_o$ (i.e., relative radius distance from the updrift control point to a point on the static planform) of this model, these authors have extended its applicability by deriving new mathematical equations to analyze the spatial and temporal changes of the eroded bay shape. These equations are not only unique but also useful for practical engineers to examining the flanking beach shape. In addition, the results of this analytical approach, are validated by one-line numerical model and shoreline video monitoring data.

**A6**). Finally, the methodology presented in this paper will benefit the readers of the Earth Surface Dynamics and related journals published by Copernicus on studying beach flanking — beach erosion at the unprotected downdrift end of a land-based coastal structure (e.g., seawall or groin), which has occurred almost universally in the field but seldom discussed in the classroom.

**B. Specific Comments**

**B1**). Now, the revised **title** is adequate to reflect the contents of the paper, which deals with the phenomenon/problem associated with beach flanking, the method used, the analysis, results and comparison, as well its application to a specific site in Korea.

For example, the original title:

"The sagging shape of shoreline formed on downdrift side of the structures due to

seasonal oblique wave incidence"

is revised as:

"Analytical approach for beach flanking downdrift of natural groin: Case study of Jeongdongjin Beach, Korea"

**B2**). The **overall structures** of this MS is now well re-organized.

For example, the original layout:
1. Introduction
2. Analysis of seasonal incident waves
   2.1 Analysis of NOAA data
   2.2 Analysis of seasonal longshore sediment transport
3. Parabolic bay shape equation
4. Analysis of shoreline change caused by oblique waves
5. Results of comparison with numerical model
   5.1 Shoreline change model
   5.2 Comparison between theoretical and numerical results
6. Results of comparison with Jeongdongjin monitoring data
7. Discussions: engineering countermeasures for mitigating seasonal erosion
8. Conclusions

are now revised as:

1. Introduction
2. Analytical approach
   2.1 Parabolic bay shape model
   2.2 Downdrift control point
   2.3 Maximum indentation point
3. NOAA's wave data for Jeongdongjin Beach
   3.1 NOAA's wave data (1979 – 2018)
   3.2 Seasonal longshore sediment transport
4. Results and comparison
   4.1 Analytical results versus numerical results
   4.2 Analytical results versus shoreline monitoring results
5. Discussions
6. Concluding Remarks

**B3**). Definition of "beach flanking" and work on beach erosion downdrift of groins are made.

Definition of "beach flanking" is now given in Abstract [L 10–13] and Introduction [L 23– 31]. Methods for analyzing beach flanking are described [L 13–20; L 60–61], and work on the beach erosion downdrift of groins are cited [L 46–57] with relevant references (see **B8)** below).

**B4**). Aims of the paper are now clearly stated [L 72–74].

**B5**). During the revision, several tasks are also involved (cited in author's esurf-2021-71-AC-

supplement):

   (1) Renumber many equations and figures, resulting from overall reorganization of sections and sub-sections;

   (2) Produce a new figure (i.e., new Fig. 1) and deleting four original figures (Figs. 5, 10, 17 and 19) and original Table 1, and

   (3) Redraw or modify eight existing figures. For example, original Figs. 4, 7, 9, 14, 15, 18, 20 and 22 are now become new Figs. 11, 13, 4, 8, 15, 10, 17 and 19.

**B6**). Limitations of the analytical approach presented in this study is stated in the Discussion [L 315– 323].

**B7**). Concluding Remark is rewritten [L362–376]

**B8**). New References relevant to groins are appended [L390–481]. For example:

Bakker, W.T. (1968).
Bakker, W.T., Klein Breteler, E.H.J., Roos, A. (1970).
Balaji, R., Kumar, S.S., Misra, A. (2017).
Dean, R.G., Dalrymple, R.A. (2002).
Hanson, H., Kraus, N.C. (1989).
Headland, J., Smith, W.G., Kotulak, P., Alfageme, S. (2000).
Hsu, J.R.C., Uda, T., Silvester, R. (2000).
Kraus, N.C., McDougal, W.G. (1996).
Larson, M., Hanson, H., Kraus, N.C. (1997).
Lehnfelt, A., Svendsen, S.V. (1958).
Longuet-Higgins, M.S. (1970a; 1970b).
Magoon, O.T., Edge, B.L. (1978).
Price, W.A., Tomlinson, K.W. (1970).
Reeve, D., Chadwick, A., Fleming, C. (2012).
USACE (2002).

**C. Recommendation**

This referee wishes to strongly recommend that the revised MS be accepted as is for publication in the Earth Surface Dynamics.

Reason: This recommendation is made based on the fact that, among the numerous papers applying the parabolic model for crenulate bay beaches since 1999, the present MS is the only one that has derived additional mathematical equations from the original parabolic bay shape equation for the phenomenon of beach flanking discussed in this MS, and the analytical results are well supported by shoreline monitoring data and numerical model using NOAA's wave data.

***** END of REPORT *****

---

## Author Response (AR2)

**Final Response to Referee 2 report for ESurf-2021-71**

[1] Title: the title is still a bit confusing (and long). Here you have some issues which are missused in the title that need revisions. Also, check their use throughout the text (e.g. abstract, introduction, discussion, conclusions).

**AC1)** The title is now revised from

"Analytical approach for beach flanking downdrift of natural groin: Case study of Jeongdongjin Beach, Korea"

to read as

"**An** analytical **model** for beach **erosion** downdrift of **groins**: Case study of Jeongdongjin Beach, Korea"

(Words in bold-face highlight the change.)

[1.1] "Flanking" is a term that it is usually, if not always, employed when describing the excess of erosion downdrift seawalls and revetments. Its use in the context of the paper will be confusing for most of readers. It will be much simpler to use something like "downdrift erosion" or similar.

**AC1.1)** As suggested, we apply the term "downdrift erosion" to replace the original "beach flanking" throughout the context (e.g. abstract, introduction, discussion, conclusions).

To avoid the likely confusion among the readers in using "beach flanking", which has occurred most commonly at downdrift of seawalls and revetments, but also downdrift of groins (as studies in this paper), we apply "beach erosion" or "downdrift erosion" in this paper when addressing the problem of erosion downdrift of groin.

[1.2] I think that authors missuse the term "Analytical approach". I assume that they are referring to the use of an analytical model to simulate shoreline erosion. However, this is not "analytical approach".

**AC1.2)** As suggested, we now use "An analytical model" rather "analytical approach" in the title and the context.

[1.3] Is the model only applicable to "natural groins"? I presume that not.

**AC1.3)** As suggested, we revise the "natural groin" to read as "groins" in title.

[1.4] A possible option could be something like "An analytical model for shoreline development downdrift of groins". You do not need to use this, but please select a more appropriate text than the current one.

**AC1.4)** Thanks for the suggestion. As shown in AC1 above, the title now reads as:

  "An analytical model for beach erosion downdrift of groins: Case study of Jeongdongjin Beach, Korea"

[2] Section 3: change the name to simply "Wave and shoreline data"

**AC3)** Thanks. We have revised the heading as suggested. [Line 214]

[3] Section 4: change the name to just "Results"

**AC4)** Thanks. We have revised the heading as suggested. [Line 279]

[3.1] I suggest to remove the subheadings 4.1 and 4.2 to present the results. Include a first sentence where you describe what results you are going to present in the chapter. State that, first, you are going to compare the performance of the analytical model with respect to the numerical model and, then the model results are compared with shoreline data.

**AC3.1)** Thanks for suggestions.

We have deleted the subheadings 4.1 and 4.2, and append a sentence at the beginning of section 4 to describe the proposed comparisons in the section, such as: [Line 280-281]

In this section, the performance of the analytical model is verified by comparing not only the numerical model

but also the shoreline data.

[3.2] Avoid to introduce new formulas or methods here. (e.g. eq 20 and 21). They should have been introduced in section 2.

**AC3.2)** As suggested, we relocate Eqs. (20) and (21) and relevant method to the end of section 2. [Line 202-213]

[4] Section 5: change the name to "Discussion"

**AC5)** Thanks. We have revised the heading as suggested. [Line 312]

[5] Section 6: change the name to "Conclusions"

**AC6)** Thanks. We have revised the heading as suggested. [Line 358]